

# Quantifying uncertainty in simulations of the West African Monsoon with the use of surrogate models

Matthias Fischer[1], Peter Knippertz[2], Roderick van der Linden[2], Alexander Lemburg[2], Gregor Pante[3], Carsten Proppe[1], and John H. Marsham[4]

[1]Institute of Engineering Mechanics, Karlsruhe Institute of Technology, Karlsruhe, Germany
[2]Institute of Meteorology and Climate Research, Karlsruhe Institute of Technology, Karlsruhe, Germany
[3]Deutscher Wetterdienst, Offenbach, Germany
[4]School of Earth and Environment, University of Leeds, Leeds, United Kingdom

**Correspondence:** Matthias Fischer (matthias.fischer@kit.edu)

**Abstract.** Simulating the West African monsoon (WAM) system using numerical weather and climate models suffers from large uncertainties, which are difficult to assess due to non-linear interactions between different components of the WAM. Here we present a fundamentally new approach to the problem by approximating the behavior of a numerical model – here the ICON (Icosahedral Nonhydrostatic) model – through a statistical surrogate model based on universal kriging, a general form of Gaussian process regression, which allows a comprehensive global sensitivity analysis. The main steps of our analysis are: (i) Identify the most important uncertain model parameters and their probability density functions, for which we employ a new strategy dealing with non-uniformity in the kriging process. (ii) Define Quantities of Interest (QoI) that represent general meteorological fields such as temperature, pressure, cloud cover and precipitation as well as the prominent WAM features Tropical Easterly Jet, African Easterly Jet, Saharan heat low (SHL) and Intertropical Discontinuity. (iii) Apply a sampling strategy with regard to the kriging method to identify model parameter combinations which are used for numerical modeling experiments. (iv) Conduct ICON model runs for identified model parameter combinations over a nested limited-area domain from 28° W to 34° E and from 10° S to 34° N. The simulations are run for August in four different years (2016 to 2019) to capture the peak northward penetration of rainfall into West Africa, and QoIs are computed based on the mean response over the whole month in all years. (v) Quantify sensitivity of QoIs to uncertain model parameters in an integrated and a local analysis. Results show that simple isolated relationships between single model parameters and WAM QoIs rarely exist. Changing individual parameters affects multiple QoIs simultaneously, reflecting the physical links between them and the complexity of the WAM system. The entrainment rate in the convection scheme and the terminal fall velocity of ice particles show the greatest effects on the QoIs. Larger values of these two parameters reduce cloud cover and precipitation, and intensify the SHL. The entrainment rate rather affects 2m temperature, 2m dew point temperature and causes latitudinal shifts, whereas the terminal fall velocity of ice mostly affects cloud cover. Furthermore, the parameter that controls the evaporative soil surface has a major effect on 2m temperature, 2m dew point temperature and cloud cover. The results highlight the usefulness of surrogate models for the analysis of model uncertainty and open up new opportunities to better constrain model parameters through a comparison of the model output with selected observations.



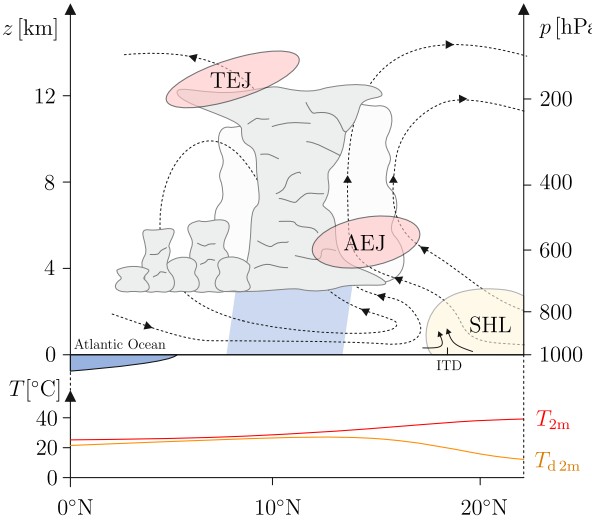

**Figure 1.** Schematic illustration of the WAM system in a height-latitude display (inspired by Fink et al., 2017) including the TEJ, the AEJ, the SHL, the ITD, 2m temperature ($T_{2m}$) and 2m dew point temperature ($T_{d\,2m}$). The main rainfall area is indicated by light blue shading. Circulation in the height-latitudinal plain is depicted through streamlines. The approximate latitudinal position of the Guinea Coast is also given.

## 1 Introduction

The West African monsoon (WAM) is a prominent seasonal large-scale circulation feature associated with a deep northward penetration of rainfall into West Africa during the boreal summer months, usually peaking in August (Hastenrath, 1991). The precipitation associated with the WAM is crucial for the livelihoods of hundreds of millions of people and has great socioeco-nomic impacts through effects on agriculture, energy production, water resources and health (Haile, 2005; Paeth et al., 2008). The WAM, conceptually depicted in Fig. 1, constitutes a complex deep overturning circulation, the formation, maintenance

and variability of which is governed by various regional and remote forcings (Hall and Peyrillé, 2006). One of its main initial drivers is the large temperature, and thus pressure gradient between the hot, dry and often dusty Sahara manifested in the Saharan heat low (SHL) and cooler, moister conditions over the tropical Gulf of Guinea. The marked discontinuity between these fundamentally different air masses, the Intertropical Discontinuity (ITD), which lies around 20° N during boreal summer, is associated with shallow and dry overturning only (Nicholson, 2009; Thorncroft et al., 2011). Abundant deep convection is

rather observed in a band south of the ITD, often called monsoonal rain belt. There, mainly between 8° and 13° N, the bulk of summertime precipitation is produced by frequently passing large convective systems with a high degree of organization (Mathon et al., 2002; Lebel et al., 2003; Lebel and Ali, 2009).

The monsoonal rain belt is enclosed by two distinctive dynamical features, the African Easterly Jet (AEJ) to the north and the Tropical Easterly Jet (TEJ) to the south. The AEJ, a pronounced easterly jet at around 600–700 hPa maintained by the low-

tropospheric meridional temperature gradient, regularly features wave disturbances. These so-called African Easterly Waves



(AEWs) with wavelengths between 2000 and 5000 km and periods of 2–7 days (Burpee, 1972; Reed et al., 1977; Kiladis et al., 2006) strongly modulate convection, mainly by enhancing vertical wind shear to levels favorable for the generation of organized squall lines (Fink and Reiner, 2003). In the upper troposphere, the WAM circulation is characterized by a jet-like intensification of the tropical easterlies. This distinct easterly current observed between 5° and 20° N, called TEJ, evolves over the South Asian monsoon system, where it is also strongest, and extends westward to Africa under gradual weakening (Flohn, 1964). Previous studies demonstrated that seasonal-mean WAM rainfall is strongly correlated with the intensity of the TEJ over West Africa (Grist and Nicholson, 2001). At least on shorter timescales, the TEJ is, however, mainly thought of as a passive feature, which can intensify after periods of increased convective activity trough the enhanced divergent outflow at upper levels (Lemburg et al., 2019).

Despite its outstanding importance for the region, simulations of the WAM spanning timescales from weather to climate remain to be fraught with substantial uncertainties. With respect to weather forecasts, Vogel et al. (2018; 2020) showed that ensemble predictions of rainfall over tropical Africa have the lowest skill throughout the tropics and are often barely better than climatological forecasts (Walz et al., 2021), even after the removal of systematic errors through statistical post-processing. This poor performance is partly related to errors stemming from initial condition uncertainty in a region known for a sparse operational network (e.g., Parker et al., 2008; Fink et al., 2011). Moreover, there appear to be issues with data assimilation, as the availability of additional observations during field campaigns shows relatively small improvements (Agustí-Panareda et al., 2010; van der Linden et al., 2020). In weather forecasts, but also in mean-state focused simulations (beyond the problem of initial state uncertainty), the representation of the WAM and its features is affected by various model uncertainties. Shortcomings in adequately simulating small-scale diabatic processes such as deep moist convection not only directly impact rainfall prediction skill but may further impose errors in the entire WAM circulation, as it is – like many tropical large-scale flows – strongly driven by the diabatic heating of the troposphere (Marsham et al., 2013; Martin et al., 2017). Model-related uncertainties regarding the representation of deep convection and other physical processes are also reflected on climate timescales where many models struggle to realistically reproduce the rainfall distribution over the WAM region and its seasonal evolution (Cook and Vizy, 2006; Xue et al., 2010; Vellinga et al., 2013). Considerable problems are also evident on paleoclimate timescales with many models struggling to accurately describe the magnitude and time of precipitation changes of the African humid period during the Holocene, which amongst other things led to a Green Sahara (Claussen et al., 2017; Brierley et al., 2020).

How can we improve model simulations over West Africa? The most obvious way is trying to improve the numerical model itself. Janicot et al. (2011) argued that biases and uncertainties can be substantially reduced if processes on weather timescales are better understood and defined. They therefore underlined the necessity of analysis on shorter timescales to not only improve weather forecasts but also climate predictions. As mentioned in the paragraph above, especially the correct representation of diabatic processes, most of them indeed acting on short time and rather small spatial scales, still constitutes a major challenge. In this regard, a key problem are model uncertainties associated with grid resolution and parameter choices in the representation of sub-grid scale processes. For example, the explicit or parameterized representation of deep convection has large impact on the amount, spatial distribution and diurnal cycle of precipitation, with substantial impacts on the large-scale dynamics and thermodynamics, even beyond the African continent (Marsham et al., 2013; Pante and Knippertz, 2019; Kendon et al., 2019).



Matsui et al. (2018) found that the treatment of radiation in their model affects precipitation, low clouds and the entire WAM circulation, while Tchotchou and Kamga (2009) highlighted the deficiencies of selected convection schemes to simulate the monsoon rainfall accurately. Gbode et al. (2018), Flaounas et al. (2011) and Klein et al. (2015) considered microphysical, convective, and boundary layer processes and found substantial influences of process parameter variations on the accuracy

and spread of precipitation and other outputs. In other studies, effects of different meteorological phenomena and boundary conditions on the WAM were investigated. For instance, Kniffka et al. (2019) highlighted that variations in low-level clouds can have a substantial impact on precipitation. Zheng and Eltahir (1998) and Hopcroft et al. (2017) investigated the influence of vegetation, where the former considered variations in the meridional distribution of vegetation on a weather timescale and the latter revealed the relationship between past vegetation coverage and climate for the mid-Holocene. Messager et al. (2004)

found that the sea surface temperature (SST) appears as a major factor in the seasonal and interannual monsoon precipitation regime.

    The above-mentioned studies were conducted to assess isolated relationships between certain model parameters and simulated WAM quantities. A general problem of this approach is that it is very challenging to study the combined effects of several sources of uncertainty at once. Non-linear interactions and buffering effects will make it nearly impossible to deduce

such effects from single-parameter perturbation experiments. Ideally one would conduct experiments across a wide range of parameter combinations but this will very quickly become too expensive, as a certain simulation period is required to separate differences from day-to-day weather noise.

    An attractive alternative to such a costly approach are surrogate models – also known as emulators or meta-models – which allow a comprehensive but resource-friendly statistical investigation of the sensitivity of QoIs to uncertain model parameters

(Cheng et al., 2020). This approach has gained increasing popularity in nearly all scientific fields, such as engineering (e.g., Sudret, 2014), chemistry and economy, within the past years (Cheng et al., 2020). For this purpose, outputs of simulations with a numerical model are used as training data to develop the surrogate models, which can then be used for a comprehensive sensitivity analysis. In meteorology, many different weather and climate models have been used for the application of surrogate models. For instance, methane emission related parameters (Müller et al., 2015) and hydrological parameters (Ray et al., 2015)

have been considered for model calibration. Fletcher et al. (2018) studied the impact of aerosol forcing and atmospheric parameters on climate sensitivity, where two cloud and convection related parameters showed the strongest impacts. Lee et al. (2011) studied the cloud condensation nuclei (CCN) sensitivity to eight emission and microphysical process parameters and found that uncertainty in the sulphur emissions explains 80 % of the output variance.

    There exist a range of methodological approaches for surrogate models. Among these, Gaussian process regression, also

known as kriging, is the most popular one in meteorological literature and has for instance been applied by Williamson (2015), Lee et al. (2011) and Fletcher et al. (2018). Alternatives include polynomial regression (Holden et al., 2009), polynomial chaos expansion (Massoud, 2019), radial basis functions (Müller et al., 2015), neural networks (Lu and Ricciuto, 2019), and combinations of these (Ray et al., 2015). For the construction, appropriate sampling strategies are used to define the training points for the surrogate model. In most cases in meteorological literature, Latin hypercube sampling (Morris and Mitchell,



1995) is used (e.g., Lee et al., 2011; Lu and Ricciuto, 2019), but in some studies other methods are applied, such as Quasi-Monte-Carlo sampling (e.g., Ray et al., 2015) and polynomial chaos based approaches (e.g., Massoud, 2019).

Universal kriging (Matheron, 1969) is a general form of Gaussian process regression, where explicit basis functions can be incorporated to describe certain relationships in the regression technique based on prior knowledge of the problem. In meteorology, universal kriging has been applied in very few studies such as by Glassmeier et al. (2019), where a linear relationship

in a two-dimensional input space is used. However, to the authors' best knowledge, universal kriging with explicit *nonlinear* basis functions has not been applied in connection with meteorological applications. Furthermore, there have not been many studies regarding criteria for the choice of basis functions for universal kriging.

This study aims at quantifying the uncertainty contributions and effects of selected model parameters on a variety of QoIs and output fields that characterize the WAM system. There has been no such study that also includes potential interactions of

multiple model parameters. The ICON (Icosahedral Nonhydrostatic) model, the operational weather prediction model of the German Weather Service (DWD), is used to simulate the rainy seasons in four years in limited-area mode. We investigate the influence of six model parameters, which are expected to have substantial impacts on the WAM characteristics. For each of them, probability density functions (PDFs) are assigned based on literature and expert knowledge. Maximin Latin hypercube sampling (Morris and Mitchell, 1995) is applied in order to define optimal parameter combinations. From the output fields of

each run, QoIs are computed that represent the characteristics of the WAM system, namely monthly accumulated precipitation, latitudinal position of the WAM rain belt, location and strength of the TEJ and the AEJ, location and extent of the SHL, latitude of the ITD and spatially averaged output fields (e.g. 2m temperature, cloud cover). Universal kriging is then used to obtain a surrogate model for each QoI, which describes the relationship between all uncertain model parameters and the QoI. The surrogate models serve to carry out global sensitivity studies and parameter studies in order to identify the parameters that

have the greatest influence on the QoIs. The results indicate for which parameters (and thus processes) uncertainties need to be reduced to lower the spread in simulated QoIs.

The paper is organized as follows: In Sect. 2, the applied methods are explained, including surrogate modeling methods and the ICON model setup. In Sect. 3, results of the conducted analyses are presented and discussed including model validation, global sensitivity analysis as well as parameter studies. Section 4 provides a summary, the main conclusions and a short outlook.

## 2  Data and method

This section details the applied methods and employed datasets. In Sect. 2.1, PDFs are assigned to considered uncertain model parameters. The surrogate modeling procedure is elaborated in Sect. 2.2, including the definition of training points, universal kriging, model validation and global sensitivity analysis. In Sect. 2.3, the ICON model setup and considered model outputs are presented. Considered QoIs and their computation are shown in Sect. 2.4. Finally, a procedure for local parameter studies is

presented in Sect. 2.5.





**Table 1.** Selected uncertain model parameters including a short description, the assumed PDF and physical unit.

| # | model parameter | ICON model parameter description | PDF | $\alpha^1$ | $\beta^1$ | unit |
|---|---|---|---|---|---|---|
| 1 | entrorg | entrainment parameter valid for dx=20 km (depends on model resolution) | log-normal | -6.3 | 0.18 | $\text{m}^{-1}$ |
| 2 | zvz0i | terminal fall velocity of ice | log-normal | 0.22 | 0.40 | $\text{m s}^{-1}$ |
| 3 | rhebc_land_trop | relative humidity threshold for onset of evaporation below cloud base over land in the tropics | beta | 30 | 10 | – |
| 4 | rcucov_trop | convective area fraction used for computing evaporation below cloud base in the tropics | log-normal | -3.0 | 0.27 | – |
| 5 | tkhmin | scaling factor for minimum vertical diffusion coefficient for heat and moisture | log-normal | -0.29 | 0.36 | $\text{m}^2 \text{ s}^{-1}$ |
| 6 | c_soil | surface area density of the (evaporative) soil surface | normal | 1.0 | 0.34 | – |

[1] parameters $\alpha$ and $\beta$ correspond to: mean $\mu$ and standard deviation $\sigma$ for normal distributions, mean $\mu$ and standard deviation $\sigma$ of the variable's natural logarithm for log-normal distributions and to respective shape parameters for beta distributions.

## 2.1 Selected uncertain model parameters

A crucial first step on the way to develop surrogate models is to identify relevant uncertain model parameters and to define meaningful PDFs representing the full epistemic uncertainty. Based on experience from sensitivity studies, literature review and expert judgement, we take into consideration six parameters which cover a fairly broad spectrum of the model's physics

such as grid-scale microphysics (*zvz0i*), turbulence (*tkhmin*), land-surface interaction (*c_soil*) and particularly the parametrization of deep convection (*entrorg*, *rhebc_land_trop*, *rcucov_trop*).

The parameter *zvz0i* designates the terminal fall speed of ice crystals, a parameter which determines the lifetime of cirrus clouds and therefore average high-level cloud cover. Particularly in the tropics, this parameter may strongly influence cloud-radiative heating rates, which can, in turn, feed back on the large-scale circulation. The choice of *tkhmin* exerts some control over the tur-

bulent diffusion of heat and moisture, which may influence cloud formation depending on static stability. The parameter *c_soil* denotes the evaporating fraction of soil in form of a unitless fraction. Higher values lead to higher relative humidity, which can possibly increase cloud cover. The parameter *entrorg* controls the mixing of ambient air into convective plumes, the so-called



entrainment. Depending on free-tropospheric humidity, higher *entrorg* values may lead to decreased buoyancy within the convective plumes and possibly reduced convective rainfall. The last two parameters concern the computation of evaporation in convective regions. The parameter *rhebc_land_trop* refers to a relative humidity threshold below which evaporation occurs below cloud base in convectively active grid cells over tropical land areas. Finally, the parameter *rcucov_trop* estimates – again specifically for the tropics – the areal fraction of convection within a grid cell that is used for the calculation of evaporation below cloud base. In contrast to *rhebc_land_trop*, which uses a threshold value for relative humidity and thus mostly affects areas where relative humidity is close to that threshold, the parameter *rcucov_trop* affects evaporation in a more general sense and thus over most of the domain. Particularly for the last three parameters within the family of convection parametrization, the net effect on area- and time-integrated precipitation is often uncertain as it strongly depends on the meteorological context.

For the purpose of the analysis in this work, the parameters are grouped into three pairs with regard to their physical implication, namely deep-cloud (*entrorg*, *zvz0i*), below-cloud (*rhebc_land_trop*, *rcucov_trop*) and boundary-layer (*tkhmin*, *c_soil*) parameters.

Despite the different physical influence of *entrorg* and *zvz0i*, the overall effects are known to be similar: Reinert et al. (2019) stated that less entrainment increases the tops of tropical convection and thus the production of cloud ice in the upper tropical troposphere. This needs to be accompanied by faster cloud ice sedimentation in order to keep the radiative forcing at a similar level. This is why the DWD varies these two parameters inversely in the ensemble physics perturbations to keep the systematic impact on the model climate small (Reinert et al., 2019).

In various meteorological studies and applications, uniform parameter distributions over an estimated range of plausible values are assumed (e.g., Wan et al., 2014), as it is also the case for the operational ensemble forecast of DWD (Reinert et al., 2019). This is reasonable if limited information is available about the considered parameters and where the main purpose of the parameter variation is to induce spread in the ensemble forecast to better reflect the full forecast uncertainty. However, in the case of a fundamental sensitivity analysis, a uniform distribution is not necessarily a good choice, as there is no physical foundation for assuming a jump in the PDF from a constant value to zero at the upper and lower limits. Therefore, other PDF choices are considered to be more appropriate in this study. Non-uniform PDFs for the parameters considered in this study have already been used by Lang et al. (2021) and Ollinaho et al. (2017), where normal and log-normal distributions are applied to represent parameter uncertainties. In our study, one source for the definition of PDFs are the mean values and ranges that are used for operational ensemble forecasts by the DWD (Deutscher Wetterdienst (DWD), 2019), including further expert knowledge. Generally, the functions are defined in such a way that physical constraints or symmetry preferences are fulfilled, e. g. parameters which are strictly positive are described by functions that can only attain positive values (e. g. log-normal PDFs) and parameters which are bounded between 0 and 1 (i. e. that describe percentages of a certain quantity) are well described by a beta distribution. The selected model parameters and the assigned PDFs are shown in Table 1. Illustrations of the PDFs are shown in Sect. 3 with the results (Fig. 5 on the bottom).



## 2.2  Surrogate modeling procedure

In order to represent the relationship between the uncertain model parameters listed in Table 1 and QoIs in a computationally effective way, surrogate models, also known as emulators or meta-models, are used. In this work, we use Gaussian process regression, also known as kriging, due to its flexibility and robustness. In order to build a surrogate model, training points (i.e., sets of combinations of the uncertain model parameters) are defined through an experimental design, and ICON model runs are conducted for these points. The individual steps necessary to develop the surrogate models are explained in the following subsections.

### 2.2.1  Training points

In order to build a surrogate model, training points for the model parameters have to be defined based on the PDFs specified in Sect. 2.1. Hereafter, we will refer to the model parameter space as *input space*, as commonly done in the scientific discipline of Uncertainty Quantification (UQ). Since probability varies strongly across the input space, it is meaningful to train the model with higher accuracy in regions with higher probability. However, defining more training points in such regions leads to an experimental design with inhomogeneous space-filling properties where surrogate modeling methods may struggle. As a consequence, the trained surrogate models may have problems to predict QoIs in the tails of the PDFs. Therefore, we transform the *physical* (hereinafter used to denote parameter PDFs according to Table 1) input space to an independent and identically distributed (i. i. d.) uniform input space. In the transformed uniform input space, which can be thought of as a multidimensional unit hypercube, every region is associated with equal probability and thus we can apply a space-filling sampling technique. In particular, we use maximin Latin hypercube sampling (Morris and Mitchell, 1995) to define 60 training points. We use the recommendation given by Loeppky et al. (2009) for choosing the number of training points as $n = 10p$, where $p$ is the number of input dimensions ($p = 6$ in our case).

For the sake of simplicity and interpretability of the results, the model parameters are kept temporarily and spatially constant during individual model runs. Therefore, one training point corresponds to a fixed set of the six model parameters which is used for one ICON model run.

The number of necessary training points strongly depends on the nonlinearity of the investigated problem. Therefore, validation (see Sect. 2.2.3) of the surrogate model remains inevitable. The obtained training points from the experimental design are transformed back into the physical input space and are then used for the configuration of the respective ICON model runs. From the outputs of the ICON model simulations for all training points, QoIs are computed as described in Sect. 2.4.

### 2.2.2  Gaussian process regression

In this study, we aim at describing a relationship between six model parameters and selected QoIs. We construct a separate surrogate model for each QoI, which can later be used to employ sensitivity studies in a resource-friendly way.

Among available surrogate modeling methods, Gaussian process regression offers wide flexibility and potential for extensions and is therefore used in this study. We apply the *universal kriging* method, a general form of Gaussian process regression,





where explicit basis functions can be incorporated. We base our choice on Fischer and Proppe (2023) where suitable basis functions for transformed input spaces have been proposed and shown to be very effective. This method is meaningful to apply in this work, since PDFs are assigned to model parameters of different orders of magnitude and the input space is thus

transformed to i. i. d. uniform random variables in order to avoid ill-conditioned problems and to apply space-filling sampling techniques.

Our aim is to build a surrogate model $\mathcal{M}$ for a QoI $y$ based on function evaluations (here: integrated quantities from ICON model simulations) $\mathbf{y} = \{y_i, i = 1 \ldots n\}$ at $n$ training points $\mathbf{X} = \{\mathbf{x}_i, i = 1 \ldots n\}$. Prediction mean and prediction variance at a set of input points $\mathbf{X}_\star = \{\mathbf{x}_{\star\,i}, i = 1 \ldots l\}$ are to be determined.

For the purpose of universal kriging,

$$g(\mathbf{x}) = f(\mathbf{x}) + \mathbf{h}(\mathbf{x})^\top \boldsymbol{\beta} \tag{1}$$

is used, with zero-mean Gaussian process $f(\mathbf{x}) \sim \mathcal{GP}(0, k(\mathbf{x}, \mathbf{x}'))$, vectors of known basis functions $\mathbf{h}(\mathbf{x}) = \{h_j(\mathbf{x}), j = 1 \ldots q\}$ and unknown coefficients $\boldsymbol{\beta} = \{\beta_j, j = 1 \ldots q\}$.

Here, the anisotropic form of the radial-basis function

$$k(\mathbf{x}, \mathbf{x}') = \theta_0 \exp\left( -\sum_{i=1}^{p} \left( \frac{|x_i - x_i'|}{\theta_i} \right)^2 \right) \tag{2}$$

with respect to hyperparameters $\boldsymbol{\theta} = \{\theta_i, i = 0 \ldots p\}$ is used as kernel to allow for different levels of smoothness between input dimensions. This makes sense, as the relationships between input parameters and QoIs are not known in advance and may differ substantially between the parameters. Furthermore, we add i. i. d. Gaussian noise with variance $\sigma_n^2$ to allow for aleatoric

uncertainties, i. e. uncertainties that are attributed to weather noise in the ICON simulations. The hyperparameters are determined by maximum likelihood estimation (Rasmussen and Williams, 2005). To speed up optimization, the gradient of the log marginal likelihood with respect to the hyperparameters $\boldsymbol{\theta}$ can be incorporated. For universal kriging, corresponding equations are given in Fischer and Proppe (2023).

Let

$$\mathbf{K} = \{K_{ij} = k(\mathbf{x}_i, \mathbf{x}_j), \qquad i = 1 \ldots n, \quad j = 1 \ldots n\},$$
$$\mathbf{k} = \{k_{ij} = k(\mathbf{x}_i, \mathbf{x}_{\star\,j}), \quad i = 1 \ldots n, \quad j = 1 \ldots l\} \quad \text{and}$$
$$\boldsymbol{\sigma}_0^2 = \{\sigma_{0\,j}^2 = k(\mathbf{x}_{\star\,j}, \mathbf{x}_{\star\,j}), \quad j = 1 \ldots l\}$$

be the vectors and matrices of kernel function evaluations and

245 $$\mathbf{H} = \{H_{ij} = h_i(\mathbf{x}_j), \quad i = 1 \ldots q, \quad j = 1 \ldots n\} \quad \text{and}$$
$$\mathbf{H}_\star = \{H_{\star\,ij} = h_i(\mathbf{x}_{\star\,j}), \quad i = 1 \ldots q, \quad j = 1 \ldots l\}$$



be the matrices of basis function evaluations at training points $\mathbf{X}$ and prediction points $\mathbf{X}_\star$, respectively.

The prediction mean and prediction variance, as shown by Rasmussen and Williams (2005), can be derived as

$$\mathcal{M}(\mathbf{X}_\star) = \mathbf{H}_\star^\top \boldsymbol{\mu} + \mathbf{k}^\top \mathbf{K}_y^{-1}(\mathbf{y} - \mathbf{H}^\top \boldsymbol{\mu}) \tag{3}$$

$$\boldsymbol{\sigma}^2(\mathbf{X}_\star) = \boldsymbol{\sigma}_0^2 - \mathbf{k}^\top \mathbf{K}_y^{-1}\mathbf{k} + \mathbf{R}^\top (\mathbf{H}\mathbf{K}_y^{-1}\mathbf{H}^\top)^{-1}\mathbf{R} \tag{4}$$

with $\boldsymbol{\mu} = (\mathbf{H}\mathbf{K}_y^{-1}\mathbf{H}^\top)^{-1}\mathbf{H}\mathbf{K}_y^{-1}\mathbf{y}$, $\mathbf{R} = \mathbf{H}_\star - \mathbf{H}\mathbf{K}_y^{-1}\mathbf{k}$ and $\mathbf{K}_y = \mathbf{K} + \sigma_n^2 \mathbb{1}$.

The selection of basis functions for universal kriging is a crucial step, because prediction accuracy of the surrogate model may strongly depend on it. Oakley (2004) emphasizes that the basis functions should be chosen to incorporate any beliefs regarding the problem, e. g. the physical evolution of the output variable depending on the input parameters. When applying universal kriging, usually low-order polynomials are used, which can often approximate physical relationships relatively well for small input parameter ranges. In addition, compared to higher order polynomials, the risk of overfitting can be reduced.

In our work, we consider transformed input spaces. Fischer and Proppe (2023) suggested using transformed basis functions to account for the input space transformation. In a general case with correlated input parameters, the Rosenblatt transformation (Rosenblatt, 1952) can be used. However, in our case we consider uncorrelated input parameters. Thus, the Rosenblatt transformation can be expressed as quantile functions (inverse cumulative distribution functions) with respect to the physical basis function of the individual model parameters (see Fischer and Proppe (2023) for more details).

In our study, we assume linear basis functions in the physical input space which are transformed into the i. i. d. uniform space by the Rosenblatt transformation. Assuming linear basis functions in the physical input space is considered reasonable, since most parameter ranges are relatively small compared to their absolute values and linear relationships may be sufficient in order to represent a global trend. Furthermore, quadratic basis functions would in a general case imply to include $n_p = p(p+1)/2 + p + 1$ basis functions, which would yield $n_p = 28$ basis functions for $p = 6$ input parameters. This number is relatively high compared to the number of training points of $n = 60$. $n$ is desired to be at least 2 to 3 times higher than the number of polynomial basis functions $n_p$ to achieve sufficiently small generalization errors (see e. g. the work by Sudret (2014) within the context of polynomial chaos expansion). The number of basis functions does not change through the transformation. Thus, we presume linear basis functions

$$\tilde{\mathbf{h}}(\tilde{\mathbf{x}}) = (1, \tilde{x}_1, \tilde{x}_2, \ldots \tilde{x}_p)^\top \tag{5}$$

w. r. t. physical input parameters $\tilde{x}_i$. By applying the Rosenblatt transformation (Fischer and Proppe, 2023), we obtain transformed basis functions $\mathbf{h}(\mathbf{x}) = \tilde{\mathbf{h}}(\mathcal{T}_{\mathrm{ros}}^{-1}(\mathbf{x}))$ w. r. t. i. i. d. uniform input parameters $x_i$. For independent physical parameters $\tilde{x}$, the vector of basis functions from Eq. (5) becomes

$$\mathbf{h}(\mathbf{x}) = (1, \mathrm{CDF}^{-1}(x_1), \mathrm{CDF}^{-1}(x_2), \ldots \mathrm{CDF}^{-1}(x_p))^\top, \tag{6}$$





where $\text{CDF}^{-1}$ is the inverse cumulative distribution function (or quantile function). This expression is used in Eq. (1) and subsequent equations.


### 2.2.3 Model validation

Obtained surrogate models need to be validated to assess their accuracy, which depends on various factors, e. g. the number of training points, the choice of basis functions and nonlinearities in the physical model. As validation criterion for a surrogate model $\mathcal{M}$ the root mean squared error (RMSE) and the normalized mean squared error (NMSE)

$$\text{RMSE} = \sqrt{\frac{1}{n_{\text{val}}} \sum_{i=1}^{n_{\text{val}}} (y_{\text{val},i} - \mathcal{M}(\mathbf{x}_{\text{val},i}))^2} \tag{7}$$

$$\text{NMSE} = \frac{1}{\sigma_{y_{\text{val}}}^2} \frac{1}{n_{\text{val}}} \sum_{i=1}^{n_{\text{val}}} (y_{\text{val},i} - \mathcal{M}(\mathbf{x}_{\text{val},i}))^2 \tag{8}$$

are used. Here, $\{\mathbf{x}_{\text{val},i}, y_{\text{val},i}, \ i=1\ldots n_{\text{val}}\}$ is a set of $n_{\text{val}}$ validation points obtained from evaluations of the numerical model and $\sigma_{y_{\text{val}}}^2$ is the variance of evaluations $y_{\text{val},i}$. Whereas the RMSE offers insight about the error in absolute values, the NMSE is a dimensionless measure which allows for better comparison between the QoIs.

Because of high computation cost, using a separate validation set that is not used for model training is not effective. Therefore, cross-validation techniques, such as leave-one-out validation or leave-k-out validation, can be applied. The validation errors for leave-one-out-validation can be formulated as

$$\text{RMSE} = \sqrt{\frac{1}{n} \sum_{i=1}^{n} (y_i - \mathcal{M}_{\setminus i}(\mathbf{x}_i))^2} \tag{9}$$

$$\text{NMSE} = \frac{1}{\sigma_y^2} \frac{1}{n} \sum_{i=1}^{n} (y_i - \mathcal{M}_{\setminus i}(\mathbf{x}_i))^2 \tag{10}$$

where $\mathcal{M}_{\setminus i}$ is the surrogate model obtained when using all $n$ training points except for the $i$th one and $\sigma_y^2$ is the variance of evaluations $y_i$. We use leave-k-out cross validation with $k=2$ as a compromise between validation accuracy and computation time. We emphasize that it is important to use generalization errors instead of measures for goodness of fit such as the coefficient of determination $R^2$, since the effect of overfitting is thereby not considered.

Model accuracy is considered to be high if NMSE values are close to $0$ and low if NMSE values are close to $1$. By definition,

values are non-negative and should not exceed $1$, as the covariance between the surrogate model and data would in that case be higher than the variance of the data itself. Interpretation of the NMSE could become problematic if QoI values do not substantially change and the variance $\sigma_{y_{\text{val}}}^2$ is very small. In such cases, the RMSE should be considered.

### 2.2.4 Global sensitivity analysis

In order to quantify the relative magnitude of dependency of the QoIs on the uncertain model parameters, global sensitivity

analysis is used. We apply FAST (Saltelli et al., 1999), a variance based sensitivity analysis, where main effect and total



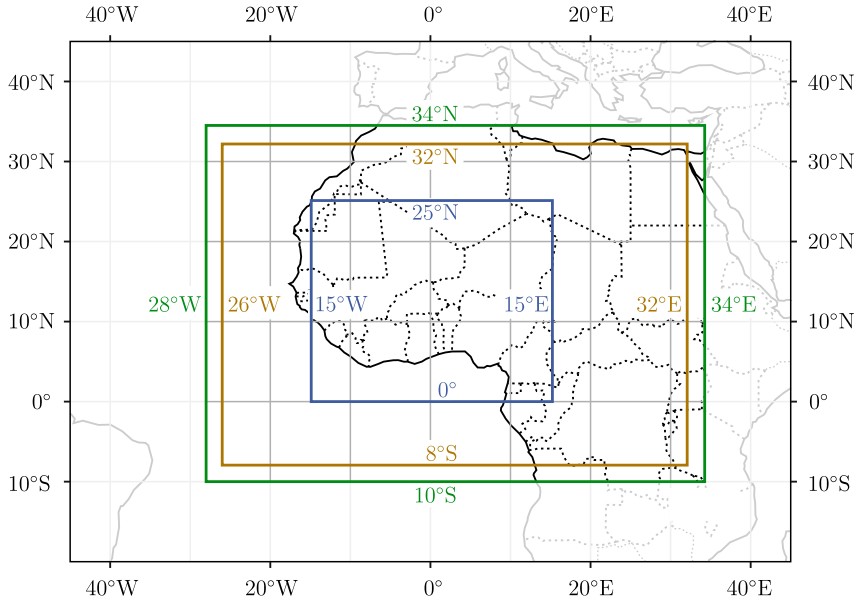

**Figure 2.** ICON model setup, outer domain with 26km grid spacing (green), inner domain with 13km grid spacing (brown) and domain for which output data are stored (blue).

effect sensitivity indices can be determined. This method has been used in many meteorological studies (e.g., Massoud, 2019; Fletcher et al., 2018). Main effect indices indicate the contribution to the output variance of varying one model parameter alone, averaged over variations in other model parameters. Total effect indices indicate contribution to the output variance of one model parameter, including all variance caused by its interactions with other model parameters. Comparison between main and total effect indices allows indication about how strong model parameter interactions contribute to the variations in the QoIs.

### 2.3 The ICON model

#### 2.3.1 Model setup

The ICON (Icosahedral Nonhydrostatic) model, the operational forecast system of the DWD, is used here as the full-physics numerical model to simulate the WAM. For this purpose, we employ the 2.5.0 model version in a limited-area nested configuration, where a 26 km grid spacing for the outer region and a 13 km grid spacing for the inner region is used. The outer area extends from 28° W to 34° E and from 10° S to 34° N with the nested domain 2° smaller in each direction as shown in Fig. 2. At the outer boundary, ERA5 reanalysis data (Hersbach et al., 2020) are used. ERA5 data is available hourly, but is updated every six hours in our simulations to limit the amount of data and computation cost. Apart from this, the model setup, including all namelist parameters, is based on the configuration used in the operational global setup by the DWD. Pante and Knippertz (2019) already obtained reasonable simulation results for the West African region with a similar model setup,





although convection parametrization turned out to be problematic for precipitation forecast. To separate sensitivities that are related to model parameters from weather noise, and to reduce the influence of initial conditions, a sufficiently long simulation period is required. At the same time we aim to fully concentrate on the peak of the WAM in boreal summer. To account for both

points, we will concentrate on August data from the four years 2016 to 2019. Each simulation starts on July 21st and is run for 41 days but only the data from August 1st to August 31st are analyzed in order to reduce initial condition influence. First tests showed that by using simulations from a single year, fluctuations in the considered QoIs were still relatively high, reflecting aleatoric uncertainties caused by small scale chaotic behavior of the atmosphere. In order to obtain a more robust measure while keeping computational cost manageable, studying rainy seasons in four years turned out to be a good compromise. All

QoIs are thus averaged over these four August periods and used as training points for the surrogate models. Using data from four different years also has the advantage of representing different states of SSTs, which are prescribed as boundary conditions and are based on the SST analysis at model initialization time. During the simulation the SST is updated incrementally based on its annual climatological cycle (Reinert et al., 2019).

### 2.3.2    Selected model output

Simulation data are stored with a horizontal resolution of 0.1° within the region from 0° N to 25° N and 15° W to 15° E (see Fig. 2) every six hours from 01 to 31 August of the years 2016, 2017, 2018 and 2019. This region and time range are hereafter denoted as *study region* and *study time*. The following model outputs were selected to represent key characteristics of the WAM:

– cloud cover at high (> 7 km), middle (2 km – 7 km) and low (< 2 km above ground) levels

– column integrated water vapor

– precipitation

– 2m temperature

– 2m dew point temperature

– mean sea level pressure

– u- and v-wind at pressure levels 200 hPa and 600 hPa

The output quantities from the ICON simulations are validated. In Sect. 2.2.3, validation of surrogate models was introduced to assess their accuracy based on given ICON model simulations, ignoring that the ICON simulations do not represent the true state of the atmosphere. For the validation in this section, the ICON model output is compared to our best estimate of the true state of the atmosphere averaged over the simulated time taken from ERA5 data. For precipitation, GPM IMERG data

(Huffman et al., 2019) is additionally included as reference. For this purpose, the ICON model output (horizontal resolution of 0.1°), native ERA5 data (horizontal resolution of 0.25°) and native GPM IMERG data (horizontal resolution of 0.1°) are



linearly remapped on a rectangular grid with a mesh size of $0.5°$. Similar to Sect. 2.2.3 the measures

$$\text{RMSE} = \sqrt{\frac{1}{n_{\text{val}}} \sum_{i=1}^{n_{\text{val}}} (y_i - y_{\text{val},i})^2} \tag{11}$$

$$\text{NMSE} = \frac{1}{\sigma_y^2} \frac{1}{n_{\text{val}}} \sum_{i=1}^{n_{\text{val}}} (y_i - y_{\text{val},i})^2 \tag{12}$$

$$\tag{13}$$

are used as validation criteria. Here, $y_i$ are averaged output values over all ICON simulations and $y_{\text{val},i}$ are corresponding ERA5 (or GPM IMERG) data at locations $i = 1 \dots n_{\text{val}}$, where $n_{\text{val}}$ is the number of grid points of the remapped grid. For the NMSE, the mean squared error is normalized with respect to the spatial variance of data $\sigma_y^2$ over the considered domain. We emphasize, that the spatial variance strongly depends on the considered domain. However, it is still used to obtain dimensionless reference

values instead of using the temporal variance, because latter may become zero in certain regions (e. g. precipitation or cloud cover in the Saharan region) and causes the measure to fail.

## 2.4 Quantities of interest

In this section, we describe the QoIs we selected to characterize the WAM and explain how these quantities are determined from the ICON model output. The results for all QoIs are averaged over the study time (01 to 31 August of the years 2016,

2017, 2018 and 2019). A schematic illustration of the monsoon system was given in Fig. 1.

**Accumulated precipitation:** The accumulated precipitation fields are computed and averaged over the study region to obtain one scalar value representing the overall precipitation.

**Precipitation latitude:** The latitude of the rain belt is determined to investigate the potential influence of model parameters on a north-south shift of the average precipitation. For this purpose, the latitudinal center of the accumulated precipitation is

computed in the study region between $12°$ W and $2°$ E (Fig. 3a) as a weighted average where the accumulated precipitation at each grid point is taken as weight for its latitude. The longitudinal range is chosen such that distinct topographical effects (Guinea highlands to the west, Cameroon line and wet Niger Delta to the east) are reduced, so the influence of model parameters on the precipitation distribution becomes more evident.

**Column integrated water vapor:** The averaged column integrated water vapor over the study region is computed.

**Cloud cover (high, mid, low):** The averaged cloud cover at high, mid and low levels over the study region is computed.

**Jet latitude:** TEJ and AEJ are the main zonal wind features of the WAM system. They are characterized by the same measures applied at different pressure levels (200 hPa for the TEJ, 600 hPa for the AEJ). The averaged latitudes of the jet streams are considered in order to investigate the potential influence of model parameters on the north-south shift of the jet streams. Computing only the latitude of the maximum zonal wind speeds turned out to be a frail measure, since it is

very sensitive and likely to fluctuate for small parameter changes due to the chaotic nature of the atmosphere. Therefore, we introduce a more robust measure, that takes the neighboring latitudes into account for computing an averaged latitude of maximum zonal wind speeds. First, we compute the averaged zonal wind speeds for each latitude on the grid. We then factorize





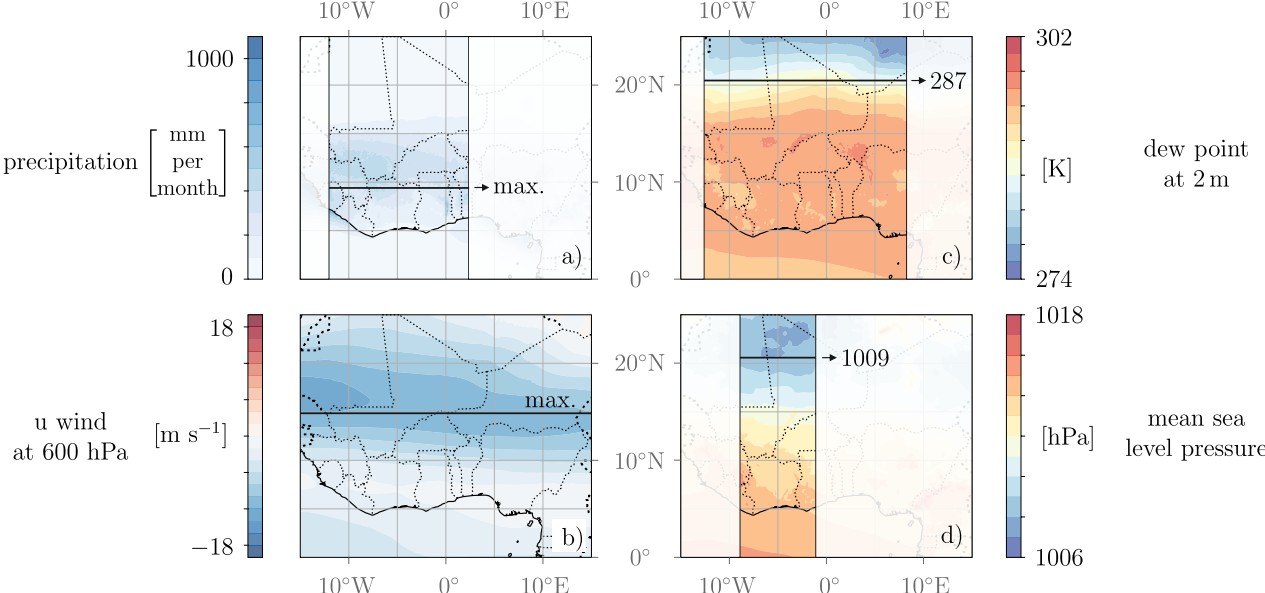

**Figure 3.** Illustrations of selected QoI computations: a) accumulated precipitation used for the determination of precipitation center latitude, b) u-wind at 600 hPa used for the determination of the AEJ latitude, c) 2m dew point temperature used for the determination of the ITD latitude based on a threshold value of 14 °C, d) mean sea level pressure used for the determination quantification of the southern boundary of the SHL based on a threshold value of 1009 hPa. All scales are linear.

the averaged wind values (here, a factor of 3 yielded useful results) to assign higher weight to the highest values and to reduce the influence of values far from the jet center which are still relatively high (Figs. 6.5 and 6.6). Finally, we determine the weighted average of latitudes analogously as for the precipitation center (Fig. 3b).

**Jet speed:** The jet speed is determined by averaging the zonal wind along the obtained jet latitude. It should be noted that the maximum jet speed at instantaneous points in time is much higher, since we consider the average of wind speeds over time for the sake of a greater robustness.

**ITD latitude:** The ITD indicates the location where dry north-easterly winds from the Sahara and moist south-westerly winds from the tropical Atlantic Ocean meet. The ITD is characterized by a marked jump in moisture content near the surface. We use the 14° C 2m dew point temperature as a measure for the ITD latitude (Fink et al., 2017). The average over latitude values between 12° W and 8° E is used as shown in Fig. 3c. Compared to the longitudinal range for the precipitation latitude, a broader range can be employed here for the sake of a more robust analysis, as the ITD behaves relatively steady.

**SHL strength:** The characteristic heat low in the region of the Sahara, one of the main drivers of the WAM, is characterized by its strength. For this purpose, the average pressure field is determined within the region from 15° N to 25° N and 15° W to 5° E, where the heat low is expected on the basis of climatological results (Lavaysse et al., 2009), for each August month. The SHL strength is characterized by the average of the 10 % lowest mean sea level pressure (MSLP) values within this region.





**SHL latitude:** The latitude of the SHL is characterized by the southern boundary of the SHL region based on a MSLP threshold of 1009 hPa between 9° W and 1° W as shown in Fig. 3d. Simulations for the investigated years indicate that the given threshold value and longitudinal range are robust measures. A broader latitude range or a higher threshold value would potentially lead to a situation where the threshold value is not reached any more for certain longitudes and the characterization would not be meaningful any more. The latitudes are computed for each monthly averaged pressure field and resulting latitudes of the four months are averaged.

**2m temperature:** The average 2m temperature over the whole study region is computed.

**2m dew point temperature:** The average 2m dew point temperature over the whole study region is computed.

## 2.5 Local parameter studies

In Sect. 2.2, surrogate modeling methods for investigating the dependency between uncertain model parameters and QoIs were introduced. However, since QoIs are defined as single values according to Sect. 2.4, information on spatial variability is lost for the benefit of robust analysis. In this section, we will therefore discuss an alternative approach to bring out the influence of uncertain model parameters on the geographical distribution of the chosen output quantities (Sect. 2.3.2). For this purpose, the same experimental design (Sect. 2.2.1) is used, but for each model parameter $i$ the training points with the 25 % lowest and the 25 % highest values $x_i$ are selected. Let these sets for each model parameter $i$ be $\mathbf{X}_{i,\text{low}}$ and $\mathbf{X}_{i,\text{high}}$. For each training point, an ICON model simulation has been conducted and output fields are available. These output fields are averaged over the whole evaluation period (four August months). Furthermore, they are then averaged over the sets $\mathbf{X}_{i,\text{low}}$ and $\mathbf{X}_{i,\text{high}}$, respectively. As a result, the spatial output data are averaged for low and high values of the considered uncertain model parameters separately. Finally, these two averaged fields are subtracted to obtain a spatial variability field. The variability plot indicates in which regions the output value becomes higher or lower for an increase of model parameter $i$.

This procedure is applied to all combinations of model parameters and available output data. To quantify the significance of such investigations, a Kruskal–Wallis test (Kruskal and Wallis, 1952) is performed. Since we cannot assume normally distributed data due to selecting 25 % training points from the tails of the distributions, a standard $t$-test is not applicable. For the Kruskal-Wallis test the difference between the data of the sets $\mathbf{X}_{i,\text{low}}$ and $\mathbf{X}_{i,\text{high}}$ from a zero field is considered. Statistical tests are conducted for every grid point and results are averaged over the whole region to include a sufficient amount of data. The test indicates whether there are significant signals in the variability fields other than random noise.

As a reference, the mean field plots can be obtained for each meteorological variable by averaging output data obtained from all available training points. These reference plots together with the variability plots can then serve as a basis for interpretation of regional influences of model parameters.





# 3   Results

## 3.1   Model validation

Validation is an essential step before discussing results of the conducted studies. It offers insight about how informative and
significant the analysis of this work is. We conducted validation for the outputs of the ICON model simulations (see Sect. 2.3.2)
and for the obtained surrogate models (see Sect. 2.2.3).

The validation results for the averaged ICON model outputs with respect to ERA5 data – and additionally GPM IMERG
data for precipitation – are shown in Table 2. For the purpose of validation, the average over the four August months on the
$0.5°$ grid is used, which should represent the climatological spatial distribution. The RMSE for precipitation is 47.7 (62.3)
mm per month for GPM IMERG (ERA5), which corresponds to 15.8 % (19.4 %) in NMSE. An inspection of the spatial
distribution shows that the differences are mostly due to wetter conditions along the rainy southwestern coast of West Africa
and Niger Delta region in ERA5 (not shown). Differences in cloudiness are also substantial. While high clouds agree best with
7.6 % RMSE, low- and mid-level cloud cover is substantially higher in ICON with RMSEs of 15.5 % and 6 %, respectively.
These correspond to NMSE of 46.9 % and 24.1 %, indicating substantial disagreement. Low clouds over tropical West Africa
are controlled by a subtle balance of advective, radiative and turbulent processes (Lohou et al., 2020), and differences between
models tend to be large (Hannak et al., 2017). Cloud cover and precipitation are strongly influenced by model parameterizations
and therefore differences are to be expected. Moreover, ERA5 itself, although much improved compared to earlier products,
may still have difficulties with these quantities (Gbode et al., 2023). The other moisture variables, i. e. column integrated water
vapor and 2m dew point temperature, however, show only minor disagreement, as does MSLP. Differences in 2m temperature
in contrast are larger (1.7 K RMSE and 20.1 % NMSE). This is mostly due to a warmer Sahara in ICON (not shown). Modelling
near-surface temperature in deserts is challenging due to the enormous solar heating and turbulent surface sensible heat fluxes,
which can lead to superadiabatic lapse rates in the lowest meters of the atmosphere (e.g., Knippertz et al., 2009). Finally, the
four wind variables show good agreement with the exception of v at 600 hPa (NMSE 26.4 %, which, however, corresponds to
an RMSE of only $0.6 \, \mathrm{m \, s^{-1}}$). This is mostly due to stronger northerlies over the Sahara in ERA5 (not shown). These validation
results show that the model setup can be generally considered to be valid, even though there are considerable differences in
certain quantities. Since in this work, sensitivity studies are conducted based on the ICON model alone, perfect agreement of
simulation output and ERA5 data is no requirement. However, for the overall significance of this work, the obtained differences
should be taken into account.

For validation of the surrogate models, leave-k-out ($k = 2$) cross-validation is applied to all QoIs individually, since separate
surrogate models have been obtained for each QoI. The RMSE and NMSE are shown in Table 3. Errors include both aleatoric
uncertainties in weather simulations (which are inevitable due to the chaotic nature of the system) and surrogate model uncer-
tainties. The prediction variance (Eq. (4)) is a measure for the uncertainty of the surrogate model. Therefore, large errors do
not necessarily mean that a surrogate model with low accuracy was obtained, but it could also mean that high aleatoric uncer-
tainties in this QoI are present. Since the magnitude of errors is robust, i. e. errors barely change for modifications and different
hyperparameters of the surrogate model, and the enhanced surrogate modeling method from Fischer and Proppe (2023) was





**Table 2.** Validation results for the ICON model outputs with ERA5 data.

| Output quantity | | RMSE | unit | NMSE |
|---|---|---|---|---|
| cloud cover (high) | | 7.60 | % | 8.80 % |
| cloud cover (mid) | | 6.03 | % | 24.14 % |
| cloud cover (low) | | 15.5 | % | 46.92 % |
| precipitation | ERA5 | 62.3 | mm per month | 19.35 % |
| | GPM IMERG | 47.7 | | 15.78 % |
| column integrated water vapor | | 1.44 | $\mathrm{kg\,m^{-2}}$ | 2.58 % |
| MSLP | | 50.7 | Pa | 4.76 % |
| temperature (2 m) | | 1.70 | K | 20.09 % |
| dew point temp. (2 m) | | 0.964 | K | 3.46 % |
| u-wind (600 hPa) | | 0.609 | $\mathrm{m\,s^{-1}}$ | 3.34 % |
| u-wind (200 hPa) | | 1.37 | $\mathrm{m\,s^{-1}}$ | 4.19 % |
| v-wind (600 hPa) | | 0.599 | $\mathrm{m\,s^{-1}}$ | 26.43 % |
| v-wind (200 hPa) | | 0.598 | $\mathrm{m\,s^{-1}}$ | 3.82 % |

applied, we presume that the aleatoric uncertainties of the model simulations dominate the RMSE (or NMSE), whereas the surrogate model uncertainty is assumed to be small. A small RMSE (or NMSE) indicates that surrogate model accuracy is high and aleatoric uncertainty is small. Small validation errors are therefore evidence that sensitivity analysis and parameter studies are meaningful. In this study, NMSEs are considered to be small for all QoIs except for the AEJ speed and precipitation

latitude. However, the small RMSEs for these quantities indicate that the absolute errors are very small. Since changes in these QoIs are very small (see also Fig. 5), the variance $\sigma_y^2$ of data used for the normalization in Eq. (12) becomes very small, too, and NMSE values become larger. Thus, large NMSE values in these cases do not affect the overall validity of this study.

### 3.2 Global sensitivity analysis

The results of the global sensitivity analysis are shown in Fig. 4. For each QoI, the bar plots indicate the main and total effect

sensitivity indices of the six uncertain model parameters. The results should be considered as relative contributions to the total variance of each QoI, such that comparison of the magnitudes between the different QoIs is not meaningful. A comparison between the absolute uncertainty contributions on the different QoIs is difficult or impossible anyway, as they have different units. Overall, the main and total effect indices do not differ strongly, which indicates that interactions between the parameters are relatively weak. This justifies interpreting influences on QoIs of individual model parameters separately as done in this

study.

Sensitivities of cloud cover (leftmost columns in Fig. 4) are generally dominated by the two deep-cloud parameters, *entrorg* and *zvz0i*. High-level clouds are strongly affected by entrainment, which can prevent convection reaching high levels,





**Table 3.** Validation results for the surrogate models of all QoIs with cross-validation.

| Quantity of Interest (QoI) | RMSE | unit | NMSE |
|---|---|---|---|
| cloud cover (high) | 0.862 | % | 1.22 % |
| cloud cover (mid) | 0.193 | % | 1.71 % |
| cloud cover (low) | 0.223 | % | 4.73 % |
| column integrated water vapor | 0.125 | $\mathrm{kg\,m^{-2}}$ | 7.04 % |
| temperature (2 m) | 0.054 | K | 3.88 % |
| dew point temp. (2 m) | 0.052 | K | 5.05 % |
| accumulated precipitation | 1.365 | mm per month | 11.64 % |
| AEJ speed | 0.069 | $\mathrm{m\,s^{-1}}$ | 55.24 % |
| TEJ speed | 0.116 | $\mathrm{m\,s^{-1}}$ | 7.02 % |
| ITD latitude | 0.100 | ° | 7.23 % |
| SHL latitude | 0.145 | ° | 4.75 % |
| AEJ latitude | 0.078 | ° | 5.81 % |
| precipitation center latitude | 0.057 | ° | 28.98 % |
| TEJ latitude | 0.044 | ° | 17.62 % |
| SHL pressure | 5.392 | Pa | 9.62 % |

in contrast to mid-level clouds, where effects are minor. The fall velocity of ice controls the dissolution of high clouds but also has a dominant effect on mid-level clouds. Low-level clouds are affected by more parameters in a more complex way. As expected, below-cloud and boundary-layer parameters have a substantial effect at these altitudes. Particularly, the parameters *rhebc_land_trop*, which affects evaporation below the cloud base, and *c_soil*, which affects surface evaporation, dominate the influence on low clouds, whereas deep-cloud parameters only play a minor role (20 % combined).

Column water vapor is mostly influenced by the deep-cloud parameters, similar to high clouds, but the boundary-layer parameters also play a minor role. This suggests that this variable is in fact more sensitive to interactions with clouds at mid- and high-levels than to changes of evaporation and vertical mixing at low levels. Somewhat unexpectedly, 2m temperature and 2m dew point temperature are mostly influenced by the deep-cloud parameters, too, with entrainment rate playing the biggest role. This suggests that these parameters must cause substantial indirect effects outside of the clouds. More obviously *c_soil* significantly affects the thermodynamics at surface level. Precipitation shows the overall most complex response being affected to various degrees by all model parameters but *rcucov_trop*. While the impact of deep-cloud parameters is no surprise, there is also a considerable impact from the boundary-layer parameters indicating the importance of low-level moisture for precipitation. The parameter *rhebc_land_trop* also shows a small influence due to the effect of evaporation below cloud base on surface rainfall.

The eight rightmost double-columns in Fig. 4 show corresponding sensitivities for the AEJ, TEJ and SHL strengths, as well as the latitude of various WAM features. The AEJ speed and position are most sensitive to *entorg* followed by *zvz0i*.



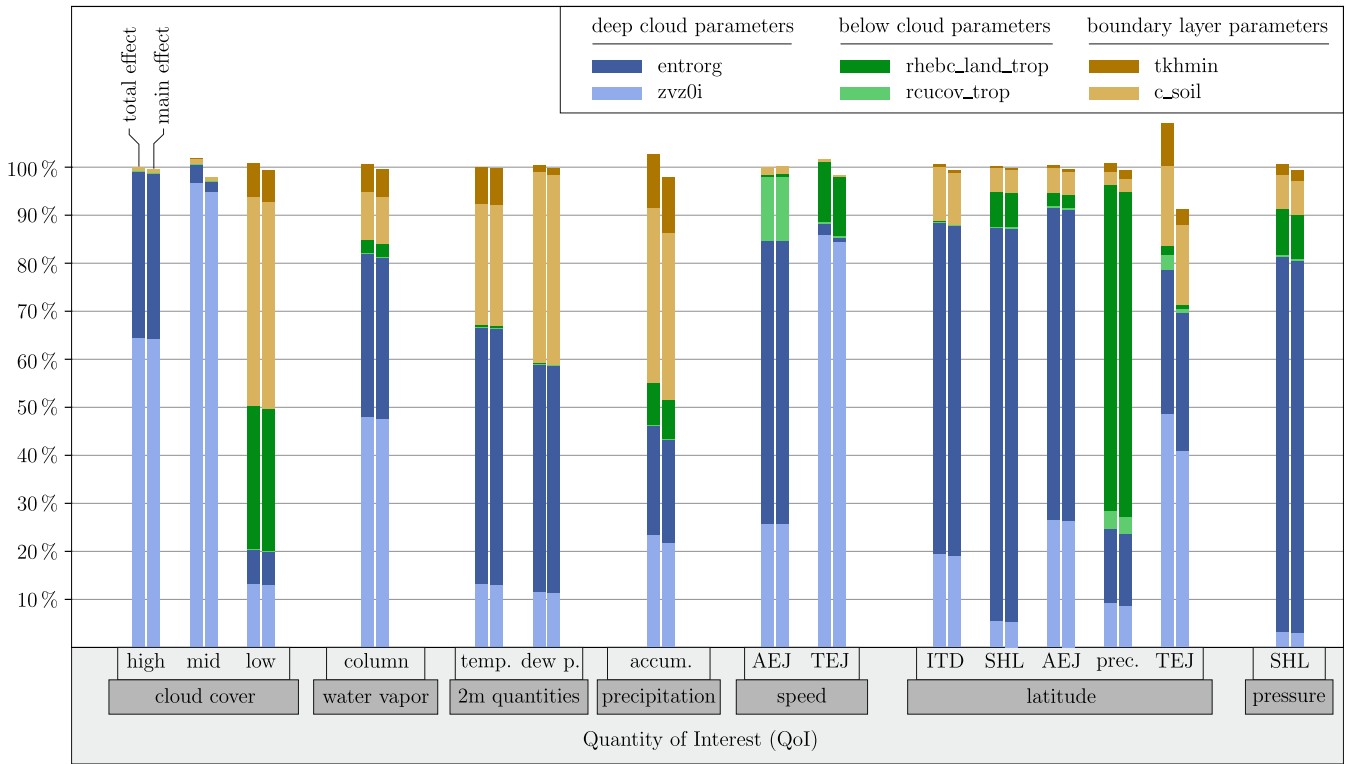

**Figure 4.** Main and total effect sensitivity indices of the six selected uncertain model parameters for all QoIs, respectively, as result of the global sensitivity analysis FAST.

This suggests that deep clouds matter most, likely through their effects on baroclinicity and vertical momentum transport. It is interesting to note that the AEJ speed is the only parameter with a considerable contribution from *rcucov_trop*, possibly due to its location in the relatively dry Sahel, where evaporation below cloud base is large. The latitudes of the ITD and the AEJ show similar sensitivities, suggesting a relatively tight coupling between the two. The TEJ speed is dominated by *zvz0i*, as this controls the divergent outflow from convective anvils, which feeds the jet (Lemburg et al., 2019). Interestingly, its position is

also sensitive to entrainment and even boundary-layer parameters, and shows the largest difference between total and main effect. The strength and latitude of the SHL are most sensitive to *entorg*, which is surprising given the absence of deep clouds over most of the Sahara. A potential explanation is that entrainment affects free-tropospheric water vapor content, which is a strong control on longwave cooling in dry regions (Pante and Knippertz, 2019). Finally, the latitude of the precipitation maximum is most sensitive to *rhebc_land_trop* ($\sim 65\,\%$ contribution) with minor contributions from all other parameters. This

behavior is in stark contrast to precipitation amount and essentially all other QoIs shown in Fig. 4. Given the large gradient in absolute and relative humidity across the Sahel, it demonstrates that shifting the onset of subcloud evaporation in the model is a powerful mechanism to shift the entire rain belt meridionally. This result may help explain some of the variability in rain belt position seen in many model intercomparison projects (e.g., Fotso-Nguemo et al., 2017).





**Figure 5.** Dependencies of all QoIs (ordinate) with respect to the six selected uncertain model parameter (abscissa), respectively. Shaded area around curves illustrates prediction variance (see Eq. (4)). In each plot, only one model parameter is varied while all other model parameters are set to their mean value. Model parameter PDFs including their mean value are shown at the bottom.



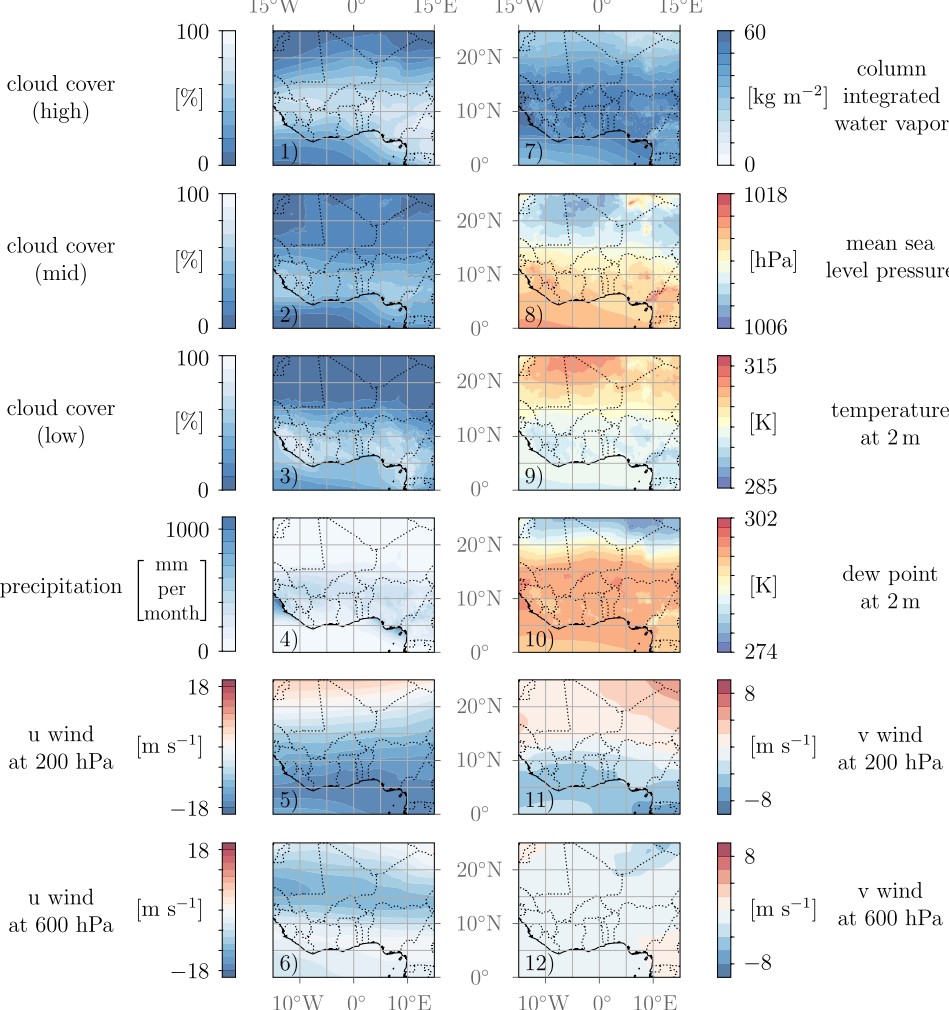

**Figure 6.** Average of selected output fields over the evaluation period (Augusts 2016–2019), averaged over all available ICON simulations. Figure numbers are chosen in accordance with the variables and labels in Figs. 7–9. All scales are linear.

### 3.3 Parameter studies

Results of the parameter studies for the QoIs based on the surrogate models described in Sect. 2.2 and of the local parameter studies described in Sect. 2.5 are discussed here consecutively for the six uncertain model parameters.

As surrogate model predictions depend on all six parameters, the full relationship cannot be visualized graphically. Instead, it is possible to illustrate one-at-a-time changes. Since parameter interactions were shown to be relatively low in Sect. 3.2, such illustrations are meaningful. Figure 5 shows the individual relationships between each model parameter and each QoI, respectively, while all other model parameters are set to their mean values. The prediction variance from the Gaussian process regression (Eq. (4)) is indicated by the shaded areas around the curves.





Averages of all output variables over all available ICON simulations and the entire evaluation period (Augusts 2016–2019), are shown in Fig. 6. Spatial variability plots for all three groups of model parameters are shown in Figs. 7, 8 and 9, respectively. The idea to compare the 25 % lowest and highest values of model parameters to investigate the regional dependencies is

supported by the fact that changes in QoIs are, if present, monotonic and in some cases even close to linear (see Fig. 5).

Results from the statistical Kruskal–Wallis test for the variability fields are denoted in Figs. 7, 8 and 9. Variability fields are denoted with two asterisks (∗∗) for very high significance (average p-value $p < 0.05$) and with one asterisk (∗) for high significance (average p-value $0.05 < p < 0.10$). Variability fields for *entrorg* and *zvz0i* are much more significant than for others. These results should be taken into account to avoid over-interpretation of non-significant cases. Significance is closely

related to the sensitivities, i. e. the greater the influence of a model parameter on a QoI (see Fig. 4), the more significant the variability field of the corresponding output quantity is in general.

### 3.3.1 Deep-cloud parameters

The effect of the investigated deep-cloud parameters, entrainment rate (*entrorg*) and the terminal fall velocity of ice (*zvz0i*) is considerably greater for most QoIs than that of other parameters, as evident from Figs. 4 and 5. Both parameters directly affect

cloudy regions only, and thus signals outside the rain belt will to some extent be due to indirect effects.

As shown in Fig. 5, the main effects of a larger *entrorg* are a decrease of 2m dew point, column water vapor, high-level cloud cover and precipitation, suggesting an overall drying of the WAM system, which is also accompanied by an increase in 2m temperature and lower pressure in the SHL. In addition, we see a consistent southward shift of the northern WAM features ITD, SHL boundary and AEJ, while the southern features, precipitation center and TEJ, remain at their latitudes. The strengths

of the jets as well as low- and mid-level cloud cover are hardly affected.

Figure 7a (i.e. 1st and 3rd column from left) shows the corresponding results on a horizontal map, which are all very highly significant according to the Kruskal-Wallis test. Increasing *entrorg* reduces precipitation to the north and south of the rain belt, as expected from Fig. 5, but surprisingly slightly increases precipitation within a narrow strip through the rain belt (Fig. 7a4). We interpret this increase as a concentration of rain to areas where ambient conditions are most suitable, while the higher

entrainment suppresses rain in more marginal areas. It is also conceivable that the southward shift of the AEJ (Fig. 5) alters the distribution of low-level wind shear, which is important for convective organization (Fink and Reiner, 2003). Despite the local precipitation increase, high clouds decrease over the entire domain by up to 25 % (Fig. 7a1) but less so over the rain belt, where they maximize climatologically (Fig. 6.1). Nevertheless, this may indicate that weaker convective systems are suppressed and that rainfall is generated more effectively by fewer, more intense systems. A higher *entrorg* also yields an increase of mid-level

clouds in the southeastern parts of the domain (Fig. 7a2), while ocean and western land areas show a slight decrease. It is generally plausible that entrainment reduces convective instability and thus retains clouds at the middle levels in marginally unstable regions but the reasons for the details of the spatial distribution are not clear. With respect to low-level cloud cover (Fig. 7a3), more entrainment implies widespread reduction over the Sahel, indicating that the northern edge of the low-cloud zone over southern West Africa (see Fig. 6.3) retreats southward, while values over the ocean and coastal areas increase. This




**Figure 7.** Spatial variability of selected output fields for the uncertain model parameters *entrorg* and *zvz0i*. The difference of the output quantity with respect to an increase of the model parameter value based on the sets $\mathbf{X}_{i,\text{low}}$ and $\mathbf{X}_{i,\text{high}}$ (see Sect. 2.5) is shown. Results of the statistical test are denoted with (∗∗): very high significance, $p < 0.05$, (∗): high significance, $0.05 < p < 0.10$. All scales are linear.

shift may be related to the overall southward shift of several WAM features, already discussed in the context of Fig. 5. The large sensitivity of high-level clouds determines the signal in total cloud cover (not shown).

The column integrated water vapor (Fig. 7a7) reduces in and around regions with less precipitation and increases (or remains the same) in wetter regions, in particular in the southeast, where we also found increased mid-level clouds (Fig. 7a2). Over the Sahara, the drying is also pronounced at the surface (2m dew point, Fig. 7a10) but less so farther south. The decrease may be a combination of less rain and evaporation plus a southward shifted monsoon circulation. The slight increases in the rain belt is probably a direct consequence of more rainfall. The overall reduced cloud cover, precipitation and thus evaporation cause a



surface warming over almost the entire land area of the domain (Fig. 7a9), associated with a lower mean-sea level pressure due to thermal expansion (Fig. 7a8), the maximum of which is to the south of the climatological SHL center (Fig. 6.8), creating a southward shift. In addition, altered temperature advection associated with the southward shift of the ITD (see Fig. 5) could

play a role.

As already pointed out in the discussion of Fig. 5, the sensitivity of the zonal jets to *entrorg* is less pronounced. The most systematic signal is the clear southward shift in zonal wind at 600hPa (Fig. 7a6) with a decrease of several $\mathrm{m\,s^{-1}}$ to the south of the climatological axis (Fig. 6.6). In the meridional direction (Fig. 7a12) we see an overall strengthening of the climatological northerlies (Fig. 6.12), indicating a stronger shallow monsoon circulation consistent with the stronger SHL (Fig. 7a8). At

the TEJ level 200 hPa the broad climatological easterlies (Fig. 6.5) are slightly weakened by larger entrainment, apart from the southeastern corner of the domain (Fig. 7a5). In the meridional direction (Fig. 7a11), the reduced rainfall over the Guinea Coast is associated with a weakening of the northerly divergent outflow towards the equatorial Atlantic (Fig. 6.11), which likely contributes to a weaker TEJ in the west (Lemburg et al., 2019). At the same time, the outflow into the northern hemisphere is slightly enhanced, shifting the relative importance of the two deep monsoonal overturning cells. Given the large decrease

in high-level cloud cover (Fig. 7a5), it is also plausible that radiative cooling in the upper troposphere increases (Stubenrauch et al., 2021), which would contribute to a weaker monsoon cell consistent with a Gill-type circulation response to a decreased off-equatorial heating (Gill, 1980).

Comparing the effect of enhanced entrainment with that of a faster terminal fall velocity of ice, we see many commonalities despite the fundamentally different microphysical processes at play. With respect to the overall effects displayed in Fig. 5,

most signals are consistent in sign (and even in magnitude). The most notable differences are a northward shift of the TEJ with higher *zvz0i*, a weaker impact on the SHL strength, a decrease in mid-level clouds and a smaller impact on the 2m dew point. Looking at the corresponding horizontal distributions (2nd and 4th column in Fig. 7) there is a striking similarity in spatial patterns, too, however with some differences in magnitude, as for example a stronger signal in high-level cloud cover (Fig. 7b1), which is directly impacted by ice particles, and weaker signals in surface temperature, dew point and pressure

as well as low-level cloud cover (Figs. 7b3,b8,b9,b10), where effects can only be indirect. The most striking difference is the absence of an anomalous behavior in the southeastern part of the study domain. Here effects of larger *zvz0i* are more consistent with other areas, i.e. implying less mid-level clouds (faster dissolution), decreased column water vapor and a weaker or unchanged TEJ (Figs. 7b2,b5,b7). Other changes in the circulation variables are almost identical (compare Figs. 7a6,a11,a12 with Figs. 7b6,b11,b12). The impact of a larger *zvz0i* on precipitation also resembles that for *entrorg* but with a smaller

amplitude.

### 3.3.2  Below-cloud parameters

The investigated below-cloud parameters, namely the relative humidity threshold for onset of evaporation (*rhebc_land_trop*) and the convective area fraction used for computing evaporation (*rcucov_trop*), also affect cloudy areas only, and thus effects outside of the rain belt will largely be indirect. Their impacts on almost all QoIs (Figs. 4 and 5) and output fields (Fig. 8) are

considerably smaller than for the deep-cloud parameters discussed in the previous subsection.



**Figure 8.** Spatial variability of selected output fields for the uncertain model parameters *rhebc_land_trop* and *rcucov_trop*. The difference of the output quantity with respect to an increase of the model parameter value based on the sets $\mathbf{X}_{i,\text{low}}$ and $\mathbf{X}_{i,\text{high}}$ (see Sect. 2.5) is shown. Results of the statistical test are denoted with ($**$): very high significance, $p < 0.05$, ($*$): high significance, $0.05 < p < 0.10$. All scales are linear.

Signals that stand out in Fig. 4 are those for low-level cloud cover and precipitation latitude (both *rhebc_land_trop*) and to a lesser extent those for precipitation amount, TEJ speed, SHL latitude and intensity (all *rhebc_land_trop*), and AEJ speed (*rcucov_trop*). Looking at the dependencies of the QoIs in Fig. 5 reveals that allowing evaporation at higher relative humidity in the model (i.e., increasing *rhebc_land_trop*) suppresses precipitation and leads to a slight southward shift of the rain belt due to a decrease in precipitation in the Sahel (Fig. 8c4), where cloud bases are higher and where subcloud relative humidity is close to the threshold climatologically. At the same time, there is a widespread increase of MSLP over the northern and central



parts of the domain (Fig. 8c8), associated with a weakening and slight northward shift of the SHL (Fig. 5). The increased subcloud evaporation is also associated with more low-level clouds over most inland areas south of the Sahara (Fig. 8c3) but 2m temperature and dew point do not show significant changes (Fig. 8c9 and c10). The small signal in near-surface temperature

could be the result of less surface evaporation due to reduced rainfall and soil moisture compensating the reduced radiative heating due to more low-level clouds and the increased subcloud evaporative cooling. Changes in temperature advection due to the weaker SHL may play a role, too. Interestingly, increasing *rhebc_land_trop* also affects high- and mid-level clouds and column water vapor (Fig. 8c1, c2, and c7) but mostly in areas away from the rain belt (i.e., Gulf of Guinea, Mauritania) and with low statistical significance. The increases in these areas are consistent with weaker overturning circulations associated

with the suppressed precipitation and possibly a redistribution of the moisture left in the atmosphere. There are some mild indications for this also in the 200hPa wind signals showing a marginally significant decrease in the northerly outflow over Nigeria (Fig. 8c11) and the strength of the TEJ (Fig. 8c5), while changes at 600hPa (Figs. 8c6 and c12) are insignificant. For *rcucov_trop*, the only field that shows significant changes is MSLP (Fig. 8d8) with a pattern similar to the signal for *rhebc_land_trop* (Fig. 8c8). In this case, the 2m temperature decrease over the Sahel (Fig. 8d9) and 2m dew point increase

over the Sahara (Fig. 8d10) are slightly more pronounced but statistically still not significant. All other fields in Fig. 8 show very weak signals, in particular the circulation and precipitation variables, consistent with Figs. 4 and 5.

### 3.3.3 Boundary-layer parameters

Effects of the scaling factor for minimum vertical diffusion for heat and moisture (*tkhmin*) and the surface area density of the evaporative soil surface (*c_soil*) are also less prominent than for the deep-cloud parameters (Sect. 3.3.1). Largest sensitivities

are found for near-surface QoIs such as low-level cloud cover, 2m temperature and 2m dew point but also for integrated quantities like column water vapor and precipitation (Fig. 4).

For a higher value of *tkhmin*, moisture is more effectively transported upwards leading to an increase in column integrated water vapor almost everywhere (Fig. 9e7, also evident in Fig. 5). Through the enhanced vertical transport of moisture, high clouds increase quite homogeneously across the domain (Fig. 9e1), while mid-level clouds increase over the rain belt but not

in a statistically significant amount (Fig. 9e2). Low-level clouds are reduced consistently over the Gulf of Guinea (Fig. 9e3), where more mixing brings drier air from the mid-troposphere into the boundary layer, supporting cloud dissolution. The mixing of drier air is also evident in lower 2m dew points in the Sahel in contrast to higher values in parts of the Sahara, where mixing may bring moister air into the boundary layer (Fig. 9e10). 2m temperature (Fig. 9e9) is hardly affected, apart from an increase over the Sahara, where longwave warming due to the higher column moisture and/or mixing of warm air from above the top

of boundary layer inversion may play a role. The enhancement of vertical exchange of moisture leads to a slight increase of accumulated precipitation (Fig. 5), which, however, is hardly visible in the spatial field (Fig. 9e4). A similar result is found for MSLP with a slight strengthening of the SHL (Fig. 5) but little signal in the spatial field (Fig. 9e8). The fact that the small mean change in MSLP is statistically significant suggests that this change is systematic without much random fluctuations.

In contrast, *c_soil* directly increases surface evaporation, leading to significantly higher 2m dew point (Fig. 9f10) and lower

2m temperature (Fig. 9f9) almost everywhere over land, which is also clearly visible in the overall dependencies shown in





**Figure 9.** Spatial variability of selected output fields for the uncertain model parameters *tkhmin* and *c_soil*. The difference of the output quantity with respect to an increase of the model parameter value based on the sets $\mathbf{X}_{i,\mathrm{low}}$ and $\mathbf{X}_{i,\mathrm{high}}$ (see Sect. 2.5) is shown. Results of the statistical test are denoted with ($**$): very high significance, $p < 0.05$, ($*$): high significance, $0.05 < p < 0.10$. All scales are linear.

Fig. 5. Column integrated water vapor (Fig. 9f7) is also enhanced but mostly to the north and south of the rain belt, where increased precipitation (Fig. 9f4, see also Fig. 5) likely removes some of the additional moisture but also slightly (and insignificantly) increases high- but not midlevel clouds (Figs. 9f1 and f2). As the increased surface latent heat fluxes over land areas moisten the boundary layer, low-level cloud cover increases over the southern parts of West Africa (Fig. 9f3), which may

further enhance near-surface cooling. This cooling in turn leads to an increased pressure (Fig. 9f8) resulting in a weakening of the SHL and northward shift of the SHL and ITD (Fig. 5).





Both *tkhmin* and *c_soil* have a remarkably similar effect on the circulation. Since an increase of the parameters yields stronger convection and more precipitation (Fig. 5), i. e. an overall strengthening of the WAM system, the 200 hPa outflow from the rain belt to the south is enhanced (Figs. 9e11 and f11) and the TEJ is accelerated (Figs. 9e5 and f5). The AEJ in

contrast is only little affected.

## 4  Conclusions

The aim of this study was to quantify uncertainty contributions of selected uncertain ICON model parameters to at set of QoIs that characterizes the WAM system. Findings should help to improve parameter specifications to make long-term simulations and forecasts more accurate. Due to computational cost, surrogate models are used as a resource friendly alternative to describe

the relationship between model parameters and QoIs. The study was based on a novel approach by Fischer and Proppe (2023) to include parameter PDFs in the construction of basis functions for universal kriging.

The dependency of QoIs on multiple model parameters and the influence of single parameters on multiple QoIs reflect the complex coupled relationships in the WAM system. Although the magnitude of the impact of individual model parameters varies quite strongly, most parameters show distinct effects on many facets of the system, which are illustrated schematically

for the four most important parameters in Fig. 10. The results can be summarized as follows:

- The entrainment rate (*entrorg*) and terminal fall velocity of ice (*zvz0i*) have the strongest effects on the WAM system (see Fig. 10a). An increase of these parameters decreases cloud cover and precipitation, mainly to the north and south of the rain belt across West Africa. Surprisingly, particularly for *entrorg*, precipitation even increases along a narrow strip through the rain belt, which may benefit from the suppressed rain elsewhere. Larger values of both parameters lead to a

stronger SHL with warmer and drier conditions in the Sahara and a stronger shallow overturning as well as a southward shift of the ITD and AEJ, while the TEJ weakens.

- The parameters *rhebc_land_trop* and *rcucov_trop* control the evaporation below the cloud base in the tropics with an overall weaker impact on the WAM. An increase of *rhebc_land_trop* (Fig. 10b) leads to less precipitation and increased low-level clouds. This appears to weaken the monsoon overturning as reflected in a weaker SHL and moister columns

in the subsidence regions over the northwestern Sahara and the Gulf of Guinea, however with little impact on AEJ and TEJ. An increase of *rcucov_trop* induces much weaker effects, particularly an increase of low-level and a decrease of mid-level cloud cover, with no substantial precipitation change.

- The scaling factor for vertical diffusion of heat and moisture (*tkhmin*) impacts on the exchange of moisture between the boundary layer and the free troposphere. An increase of this parameter therefore increases column water vapor and leads

to more high- and mid-level clouds but precipitation is hardly affected. The evaporative soil surface (controlled by *c_soil*) also increases column water vapor and cloud cover, but in this case mainly the low-level clouds, even leading to a small increase in precipitation at the southern side of the rain belt (see Fig. 10c). Near-surface temperature decreases through





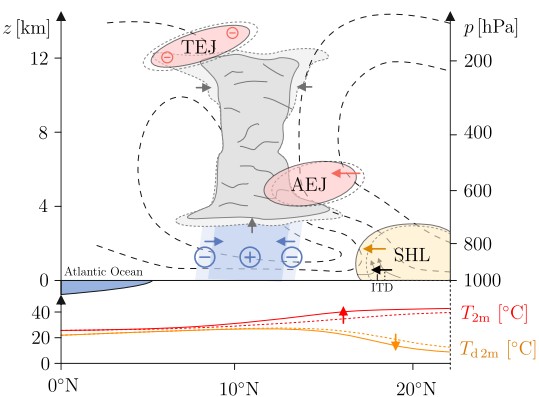

(a) Deep-cloud parameters (*entrorg*, *zvz0i*)

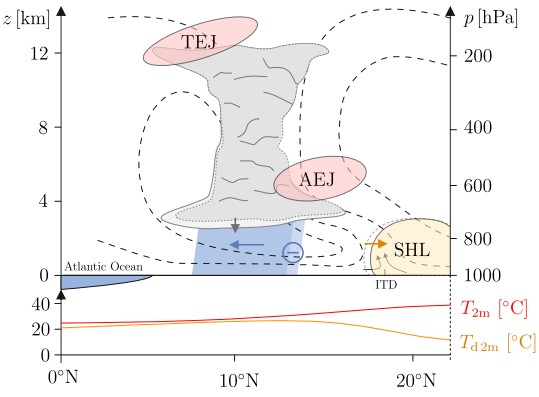

(b) Below-cloud parameters (*rhebc_land_trop*)

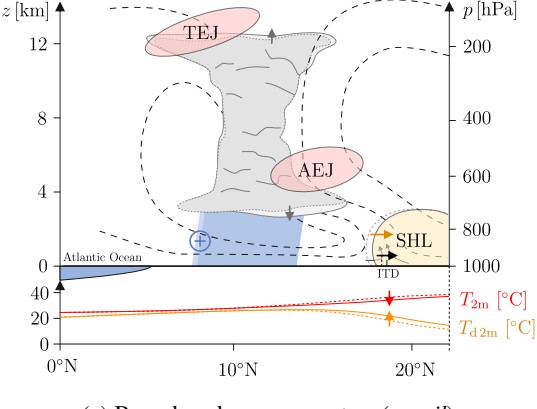

(c) Boundary-layer parameters (*c_soil*)

**Figure 10.** Illustration of qualitative effects on the WAM system by an increase of the investigated uncertain model parameters according to the panel captions. See Sect. 3 for a more detailed discussion.



increased evaporation, while 2m dew point temperature and MSLP increase shifting the SHL northwards. Impacts on the AEJ and TEJ are rather small for both *tkhmin* and *c_soil*.

Concerning the selected uncertain model parameters (Sect. 2.1), given limited information from literature, the definitions are rough estimates and obtained results should be interpreted with some caution. Furthermore, only six model parameters are included in the study, but other parameters may also have relevant uncertainty contributions. Moreover, the results based on the used ICON model version cannot directly be transferred to other model versions or even other models, where different parameters are used in parameterizations. Nevertheless, the outcome of this study highlights the usefulness of the applied

methodology including training procedure and surrogate models. The methodology is not limited to few model parameters, but can be extended. The computational effort is expected to increase linearly with the amount of model parameters (see Sect. 2.2.1). As this study has shown that the entrainment rate has a strong influence, other related parameters might be of interest, such as distinction between turbulent and organized entrainment as well as detrainment parameters. Another interesting parameter for future studies might be cloud inhomogeneity.

This study has shown that it is mainly the entrainment rate, the fall speed of ice and surface evaporation that should be specified more accurately. This can be done by including further investigations, measurements and expert knowledge including a more complex representation in parameterizations. Moreover, these parameters could be optimized with respect to the WAM simulation through parameter identification studies by including reanalysis and satellite data as observational references. The surrogate models that were obtained in this study can serve as the basis to conduct such identification studies. However, the

outcome would be limited to the West African region. Thus, it might be possible, to specify parameters that should only be valid in regions for which they have been optimized, as it is already the case for *rhebc_land_trop* and *rcucov_trop*, which have been tuned for tropical regions. The implementation of parameter identification studies based on the obtained surrogate models is currently ongoing.

*Author contributions.* PK conceived the overall concept of the study including all necessary steps to quantify uncertainties of selected
model parameters. MF designed the study including experimental design, surrogate models, computation of QoIs, sensitivity analysis and local parameter studies with input from all co-authors. GP set up the ICON model including ERA5 data. RVDL, GP and PK supported with meteorological aspects in the model setup. CP supported with methodical aspects of the study. PK, AL and JM contributed with interpretation of results and strategies for post-analysis. MF prepared the manuscript with input from all co-authors.

*Competing interests.* PK is a member of the editorial board of WCD.

*Acknowledgements.* PK acknowledges project C2 "Prediction of wet and dry periods of the West African Monsoon" of the Transregional Collaborative Research Center SFB/TRR 165 "Waves to Weather" funded by the German Science Foundation (DFG).



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
