# Peer review of "Quantifying uncertainty in simulations of the West African Monsoon with the use of surrogate models"

_EGUsphere, 2023_

## Author Response (AR1)

**Point-by-point response to referee comments**

**Quantifying uncertainty in simulations of the West African Monsoon with the use of surrogate models**

*Matthias Fischer, Peter Knippertz, Roderick van der Linden, Alexander Lemburg, Gregor Pante, Carsten Proppe, John H. Marsham*

We would like to thank the reviewers for their constructive and helpful comments on the manuscript. Overall, we agree with the given remarks and provide a short response below. For other minor (technical) comments that are not mentioned below, we do not provide responses here but will modify respective parts of the manuscript.
* * *
**- Line 19-20:** '…rather affects…' This sounds weird and is unclear. Please clarify.

*will be clarified*

> the SHL. The entrainment rate  primarily affects 2m temperature, 2m dew point temperature and causes latitudinal shifts,
>
> 20   whereas the terminal fall velocity of ice mostly affects cloud cover. Furthermore, the parameter that controls the evaporative soil
* * *
**- Line 96:** '…within the past years…' This is vague (what timescale is 'the past'?) and needs rephrasing. Maybe '…within the last XX years…', or, '…within recent years…'?

*will be clarified*

> 95   allow a comprehensive but resource-friendly statistical investigation of the sensitivity of QoIs to uncertain model parameters
>
> (Cheng et al., 2020). This approach has gained increasing popularity in nearly all scientific fields, such as engineering (e.g.,
>
> Sudret, 2014), chemistry and economy, within the past  three decades (Cheng et al., 2020). For this purpose, outputs of
>
> simulations with a numerical model are used as training data to develop the surrogate models, which can then be used for a
* * *
**- Line 114:** '…In meteorology, universal kriging has been applied in very few studies…'. I don't agree with this. I think a version of universal kriging has been applied in several studies that relate to meteorology, including tropical sea breeze convection (e.g. 10.1029/2019JD031699) and deep convective clouds and hail (e.g. 10.5194/acp-20-2201-2020). Please amend to reflect this.

*Universal kriging indeed has been applied in a few studies. The proposed references by both reviewers (Diamond, M. S. (2020), Wellmann (2020)) are good examples which we will include in our literature review.*

*However, in J. M. Park (2020) no background is given about the universal kriging method and whether/how/why it is applied rather than using simple or ordinary kriging. Therefore, we decided not to include it in the context of universal kriging.*

> 115 meteorology, universal kriging has been applied in  several studies such as by  Glassmeier et al. (2019); Wellmann et al. (2020); Diamond et al. (2020), where either linear or quadratic basis functions are used as trend functions. However, to the authors' best knowledge, universal kriging with explicit *nonlinear* basis functions other than polynomials has not been applied in connection with meteorological applications. Furthermore, there have not been many studies regarding criteria for the choice of basis functions for universal
> 120 kriging.

- **Line 134:** '…as well as parameter studies.' – What do you mean by 'parameter studies'. Is this not the sensitivity analysis? Please explain / clarify.

*will be clarified*

> QoI. The surrogate models serve to carry out  sensitivity and parameter studies. Here, the global sensitivity analysis (GSA) evaluates the quantitative influence of the PDFs of all model parameters on the variability of the QoIs, whereas the parameter studies involve varying one
> 135 parameter at a time to observe the relationship between the parameter and each QoI. The results indicate for which parameters (and thus processes) uncertainties need to be reduced to lower the spread in simulated QoIs.

- **Table 1:** I find the way the parameters are presented in Table 1 quite confusing. Only one distribution is Beta, and yet the parameter columns are labelled as beta parameters as a default? It would be much easier to understand if the distributions were just written in full in a single 'PDF' column, e.g. Normal(1.0, $0.34^2$) or Normal($\mu$=1.0, $\sigma^2$=$0.34^2$), and I recommend doing this.

*We agree that this labeling, which was chosen to keep the notation plain and compact, may be misleading. We will modify the notation according to the suggestion.*

**Table 1.** Selected uncertain model parameters including a short description, the assumed PDF and physical unit.

| # | model parameter | ICON model parameter description | PDF[1] | unit |
|---|---|---|---|---|
| 1 | entrorg | entrainment parameter valid for dx=20 km (depends on model resolution) | Lognormal $(\mu = -6.3, \sigma = 0.18)$ | $\text{m}^{-1}$ |
| 2 | zvz0i | terminal fall velocity of ice | Lognormal $(\mu = 0.22, \sigma = 0.40)$ | $\text{m s}^{-1}$ |
| 3 | rhebc_land_trop | relative humidity threshold for onset of evaporation below cloud base over land in the tropics | Beta $(\alpha = 30, \beta = 10)$ | – |
| 4 | rcucov_trop | convective area fraction used for computing evaporation below cloud base in the tropics | Lognormal $(\mu = -3.0, \sigma = 0.27)$ | – |
| 5 | tkhmin | scaling factor for minimum vertical diffusion coefficient for heat and moisture | Lognormal $(\mu = -0.29, \sigma = 0.36)$ | $\text{m}^2\ \text{s}^{-1}$ |
| 6 | c_soil | surface area density of the (evaporative) soil surface | Normal $(\mu = 1.0, \sigma = 0.34)$ | – |

[1] for log-normal distribution: $\mu$ and $\sigma$ are the mean and standard deviation of the variable's natural logarithm.
* * *
 **- Line 143:** '…define meaningful PDFs representing **the full epistemic uncertainty**.' Is this possible? Is the 'full epistemic uncertainty' actually known. [epistemic uncertainty is uncertainty due to a lack of knowledge – so, this includes the 'unknown unknowns' part as well as the 'known unknowns' – Hence, I think it may be more correct to say that these PDFs will contain our best knowledge of the associated epistemic uncertainty, rather than the *full* epistemic uncertainty. Please amend appropriately.

*Thank you for this comment - this is absolutely correct and we will adapt the formulation accordingly.*

A crucial first step on the way to develop surrogate models is to identify relevant uncertain model parameters and to define meaningful PDFs representing  our best knowledge of the associated epistemic uncertainty. Based on experience from
150  sensitivity studies, literature review and expert judgement, we take into consideration six parameters which cover a fairly broad
* * *
**- Lines 144-145:** For clarity, change '…physics, such as grid-scale…' to '…physics. These are the grid-scale…'. 'such as' suggests there may be other options, but the following list contains all of the parameters considered. Also, at the end of the sentence (L 146), please add a reference to Table 1.

*will be clarified*

150    sensitivity studies, literature review and expert judgement, we take into consideration six parameters which cover a fairly broad

spectrum of the model's physics. These are the grid-scale microphysics (*zvz0i*), turbulence (*tkhmin*), land-surface inter-

 - **Line 160:** 'Particularly for the last three parameters within the family of convection parametrization' – This does not read well and needs clarification. Which parameters are being referred to here? Also, the order of the parameter descriptions in the text does not correspond to the order they are listed in Table 1, which may confuse the reader – please consider aligning these orders.

*will be clarified. For the order and grouping of parameters, we will adapt the description according to the Table 1 and Result section.*

action (*c_soil*) and particularly the parametrization of deep convection (*entrorg*, *rhebc_land_trop*, *rcucov_trop*). For the purpose of the analysis in this work, the parameters are grouped into three pairs with regard to their physical implication, namely

deep-cloud (*entrorg*, *zvz0i*), below-cloud (*rhebc_land_trop*, *rcucov_trop*) and boundary-layer (*tkhmin*, *c_soil*) parameters (see Table 1).

The  entrainment rate (*entrorg*) controls the mixing of ambient air into convective plumes. Depending on the free-tropospheric humidity, higher *entrorg* values may lead to decreased buoyancy within the convective plumes and possibly reduced convective rainfall. The terminal fall speed of ice crystals  (*zvz0i*) determines the lifetime of cirrus clouds and therefore average high-level cloud cover. Particularly in the tropics, this parameter may strongly influence cloud-radiative heating rates, which can, in turn, feed back on the large-scale circulation. ~~The choice of tkhmin exerts some control over the turbulent diffusion of heat and moisture, which may influence cloud formation depending on static stability. The parameter c_soil denotes the evaporating fraction of soil in form of a unitless fraction. Higher values lead to higher relative humidity, which can possibly increase cloud cover. The parameter entrorg controls the mixing of ambient air into convective plumes, the so-called entrainment. Depending on free-tropospheric humidity, higher entrorg values may lead to decreased buoyancy within the convective plumes and possibly reduced convective rainfall. The last two~~ Despite the different physical influence of the entrainment rate and the terminal fall velocity of ice, the overall effects are known to be similar: Reinert et al. (2019) found that less entrainment increases the tops of tropical convection and thus the production of cloud ice in the upper tropical troposphere. This needs to be accompanied by faster cloud ice sedimentation in order to keep the radiative forcing at a similar level. This is why the DWD varies these two parameters inversely in the ensemble physics perturbations (a probabilistic forecast where model parameters are varied to generate a range of possible outcomes to account for uncertainties in the model's physics) to keep the systematic impact on the model climate small (Reinert et al., 2019). The below-cloud parameters concern the computation of evaporation in convective regions. The parameter *rhebc_land_trop* refers to a relative humidity threshold below which evaporation occurs below cloud base in convectively active grid cells over tropical land areas.  The parameter *rcucov_trop* estimates – again specifically for the tropics – the areal fraction of convection within a grid cell that is used for the calculation of evaporation below cloud base. In contrast to *rhebc_land_trop*, which uses a threshold value for relative humidity and thus mostly affects areas where relative humidity is close to that threshold, the parameter *rcucov_trop* affects evaporation in a more general sense and thus over most of the domain.

The choice of *tkhmin*

 influences the turbulent diffusion of heat and moisture, which can in some situations impact cloud formation. Some level of vertical diffusion is practically always present in reality. In case of highly stable conditions and weak vertical wind shear, however, this diffusion is underestimated by the turbulence parametrization. Therefore a minimum vertical diffusion (i. e., *tkhmin*) is set in the model to counteracts this underestimation (Raschendorfer, 2012). The parameter *c_soil* denotes the evaporating fraction of soil in the form of a unitless fraction. Higher values lead to higher relative humidity, which can possibly increase cloud cover. Particularly for *entrorg*, *rhebc_land_trop* and *rcucov_trop*, the

net effect on area- and ~~zvz0i, the overall effects are known to be similar: Reinert et al. (2019) stated that less entrainment increases the tops of tropical convection and thus the production of cloud ice in the upper tropical troposphere. This needs to be accompanied by faster cloud ice sedimentation in order to keep the radiative forcing at a similar level. This is why the DWD varies these two parameters inversely in the ensemble physics perturbations to keep the systematic impact on the model climate small (Reinert et al., 2019).~~ time-integrated precipitation is uncertain, as it strongly depends on the meteorological context.
* * *
 **- Line 165:** '…of *entrorg* and *zvz0i*…' Here and elsewhere, I find it difficult to remember the descriptions of the parameters from the model names/acronyms, which don't seem intuitive to me. I think it would help to be more descriptive in the text, e.g. '…of the entrainment and

terminal ice velocity parameters, *entrorg* and zvz0*i*,…' so the reader doesn't have to keep looking things up.

*will be clarified*

> The  entrainment rate (*entrorg*) controls the mixing of ambient air into convective plumes. Depending on the free-tropospheric humidity, higher *entrorg* values may lead to decreased buoyancy within the convective plumes and possibly reduced convective rainfall. The terminal fall speed of ice crystals  (*zvz0i*) deter-
> mines the lifetime of cirrus clouds and therefore average high-level cloud cover. Particularly in the tropics, this parameter may
> 160   strongly influence cloud-radiative heating rates, which can, in turn, feed back on the large-scale circulation.
* * *
**- Line 168:** '…in the ensemble physics perturbations…'. What are these? This needs more explanation.

*will be clarified with better reference to the given literature source*

> the radiative forcing at a similar level. This is why the DWD varies these two parameters inversely in the ensemble physics
> 170   perturbations (a probabilistic forecast where model parameters are varied to generate a range of possible outcomes to account
> for uncertainties in the model's physics) to keep the systematic impact on the model climate small (Reinert et al., 2019). The
> below-cloud parameters concern the computation of evaporation in convective regions. The parameter *rhebc_land_trop* refers

> Reinert, D., Prill, F., M., D., and Zängl, G.: Database Reference Manual for ICON and ICON-EPS, Version 1.2.11, Tech. rep., 2019.
* * *
**- Lines 174-175:** '…in the case of a fundamental sensitivity analysis, a uniform distribution is not necessarily a good choice, as there is no physical foundation for assuming a jump in the PDF from a constant value to zero at the upper and lower limits.' I'm not sure I fully agree here. For sensitivity analysis, uniform distributions are used to reflect that (under current knowledge) a parameter's value is equally likely to be any value across a given range. Beyond that range (and so the physical meaning of it) is irrelevant to the sensitivity analysis, as it doesn't analyze beyond those limits.

In terms of physical foundations for the distribution choices that are used in this study (Table 1, Figure 5, what is the 'physical foundation' for the shapes and rates of decay in the tails of the PDFs selected? – How exactly were these distributions chosen? (Was there a robust elicitation process?) And, how realistic are they?

The appropriateness of non-uniform distributions can also be questioned – When multiple peaked marginal PDFs come together this can highly bias the sampling of a multi-dimensional parameter space and effectively places a strong constraint on that parameter space prior to any actual calibration. How confident are the authors in the accuracy of these distribution specifications and the constraints to their analysis that these PDFs impose?

*We understand and emphasize that assigning PDFs to the parameters is a crucial and important step of the whole analysis, which is by no mean trivial. The selected PDFs do have a direct impact on the results of the global sensitivity analysis, but not on the parameter studies, where the relationship between the QoIs and physical parameters are shown. For the global sensitivity analysis, we may ask the question whether a uniform or a more sophisticated PDF choice is more meaningful. Here, we concluded that a uniform distribution with equal probabilities within a certain range and zero probability beyond the limits would be rather dubious, because parameter values close to the limits within the range would contribute to the global sensitivity analysis with 'full' weight and parameter values beyond the limit (but still close to the range) with zero weight. Although a uniform distribution would not be the best choice in our opinion, defining other distributions is challenging. We already elaborate our choices in L178-184 but will expand this and add that other definitions, e.g. uniform distribution are often preferred by other authors.*

the parameter variation is to induce spread in the ensemble forecast to better reflect the full forecast uncertainty. Since the definition of PDFs is often challenging and vague, we emphasize that using uniform ranges is often preferable. However, in

200   the case of a fundamental sensitivity analysis, a uniform distribution is not necessarily a good choice, as there is no physical foundation for assuming a jump in the PDF from a constant value to zero at the upper and lower limits.  Values near the uniform range limits would have a disproportionate influence in the global sensitivity analysis, while those just outside the range would contribute nothing. Although defining alternative distributions presents a challenge in its own right, these are considered to be more appropriate in this study. Non-uniform PDFs for the parameters considered in this

205   study have already been used by Lang et al. (2021) and Ollinaho et al. (2017), where normal and log-normal distributions are applied to represent parameter uncertainties. In our study, one source for the definition of PDFs are the mean values and
* * *
**- Lines 195-196:** 'Since probability varies strongly across the input space, it is meaningful to train the model with higher accuracy in regions with higher probability.' I'm not sure I fully understand the logic of this… just because the probability distributions suggest you may not sample an area of parameter space as frequently as another (i.e. in a sensitivity analysis), does that mean that you should want or accept higher error in the predictions (and so less-informed predictions) when you do sample there? Could having variable errors in prediction accuracy across the parameter space lead to bias in the results (e.g. sensitivity analysis) from the sampling (even with lower frequency of samples) of the areas (edges of parameter space) with lower probability / lower accuracy? Has this been tested?

In my mind, one should want the surrogate model (emulator) to be as good a representation as possible of the complex model (simulator) across all of the parameter uncertainty space considered, to then be confident in using that representation in place of the simulator for all sampled parameter combinations.

I think to take this approach of weighted emulator accuracy, you need to be highly confident in the accuracy of the parameter PDFs being used to create that weighting (connects to comment above). However, in the conclusion (Lines 670-671) you suggest this is not the case. Also, other factors such as the smoothness of the output response surface can affect the number of parameter combinations required to obtain a reasonable emulator (a rougher response surface may require more training information) – Would a rougher surface in an

area of lower probability exacerbate the potential bias in results from prediction accuracy in such a weighted approach?

I'm interested to know how different the results would be if the training data were sampled evenly over the physical parameter uncertainty space without the PDFs – This would indicate the need/benefit, or not, for this more complex and weighted sampling approach.

*We create the surrogate models in order to carrying out global sensitivity analyses. The amount/density of points in the parameter space which are used for conducting the sensitivity analysis corresponds to the probability distribution. This means that in order to get a more precise sensitivity analysis, it is meaningful that the model is more accurate in regions in the parameter space with higher PDF values. If the only aim is to construct a surrogate model that should be just as accurate in the 'tails' of the parameters, then we would concede that the reviewer's comment is correct. However, a comparison study using the meteorological model is difficult: to do this, the entire ICON model runs would have to be carried out again with different parameter combinations that were sampled differently. Furthermore, we suspect that the results would not differ much. In our opinion, our approach is the more intuitive/elegant approach and also optimal in terms of the global sensitivity analysis.*

*A comparative study may be subject to future research using less expensive toy/academic problems in a rather methodical/mathematical paper. It would indeed be interesting to investigate whether rough model behavior in the tails could lead to lower overall accuracy or even biases.*

> of Uncertainty Quantification (UQ). Since probability varies strongly across the input space, it is meaningful to train the model with higher accuracy in regions with higher probability.  This is because we construct surrogate models
> 225 particularly for performing global sensitivity analysis. The density of points selected in the parameter space for this analysis corresponds to the probability distribution. Thus, for a more accurate sensitivity analysis, it is crucial for the model to exhibit higher accuracy in regions of the parameter space where the PDF values are greater. However, if the only aim was to develop a surrogate model with equal accuracy across the whole parameter space including the tails of the PDFs, then using a uniform density of training points would be more suitable. In our case, using more training points in  regions with higher
> 230 probability leads to an experimental design with inhomogeneous space-filling properties where surrogate modeling methods may struggle. As a consequence, the trained surrogate models may have problems to predict QoIs in the tails of the PDFs.
* * *
 **- Line 214:** '…be used to employ sensitivity studies in a resource-friendly way.' What is meant by 'resource-friendly'? Please clarify.

*will be clarified*

245 **2.2.2 Gaussian process regression**

In this study, we aim at describing a relationship between six model parameters and selected QoIs. We construct a separate surrogate model for each QoI, which can later be used to employ sensitivity studies with significantly reduced computational cost.
* * *
**- Line 234:** 'Furthermore, we add i. i. d. Gaussian noise with variance sigma_n^2…'. It isn't clear how this is done. Please clarify.

*will be clarified (see comment below L459ff)*

with $\boldsymbol{\mu} = (\mathbf{H}\mathbf{K}_y^{-1}\mathbf{H}^\top)^{-1}\mathbf{H}\mathbf{K}_y^{-1}\mathbf{y}$, $\mathbf{R} = \mathbf{H}_\star - \mathbf{H}\mathbf{K}_y^{-1}\mathbf{k}$ and $\mathbf{K}_y = \mathbf{K} + \sigma_n^2\mathbb{1}$.

285 Here, additive i. i. d. Gaussian noise with variance $\sigma_n^2$ is considered, where $\sigma_n^2$ is treated as another hyperparameter to allow for aleatoric uncertainties, i. e. uncertainties that are attributed to weather noise in the ICON simulations.  In
* * *
**- Line 314:** Is there a general reference for the ICON model, for if a reader wants to find out more details?

*We referred to one version of the ICON manual (Reinert D., 2019) but we will revise this again and add a reference where introducing the ICON model in the manuscript.*

**2.3.1 Model setup**

The ICON (Icosahedral Nonhydrostatic) model (Zängl et al., 2015), the operational forecast system of the DWD, is used here as the full-physics numerical model to simulate the WAM. For this purpose, we employ the 2.5.0 model version in a limited-

Zängl, G., Reinert, D., Rípodas, P., and Baldauf, M.: The ICON (ICOsahedral Non-hydrostatic) modelling framework of DWD and MPI-M: Description of the non-hydrostatic dynamical core, Quarterly Journal of the Royal Meteorological Society, 141, 563–579,
970    https://doi.org/10.1002/qj.2378, 2015.
* * *
**- Line 330:** '…QoIs are thus averaged over these four August periods…' Does the averaging over the 4 years lead to an overall behavior that is still realistic? (i.e. Is it possible that for a process, the different meteorology might lead to a high value or a low value, but then the averaging leads to a more central value that is never observed?)

*In a preliminary study, we only included one August period and found that the fluctuations in the relationship between parameters and outputs were relatively large. Therefore, we averaged over 4 months (always August to cover similar climatology). Due to the fluctuations for individual years, it was not possible to investigate the differences between the years with*

*sufficient significance. However, the fact that we get a more robust signal by using four months (significant results in the model validation) strongly suggests that we obtain a smoothing rather than a cancellation of the individual signals. Thus, we are confident to have a realistic estimate of the averaged behavior.*

370  models. Preliminary studies with only one August month were still relatively volatile in a sense of large uncertainties in the surrogate model, which could then be strongly reduced by including four months. Validation of the surrogate model is essential to ensure that averaging over four years does not lead to non-realistic behavior in the average. The results of this validation should demonstrate a robust signal, indicating that the process results in a smoothing of individual signals rather than their cancellation. Using data from four different years also has the advantage of representing different states of SSTs, which are
* * *
**- Line 335:** Is it possible to give an indication of the actual amount/size of data that is stored (required level of storage for if someone wanted to repeat this).

*will be added*

390    The data necessitate approximately 475 gigabytes of storage space.
* * *
**- Lines 339-345:** Please add units to all of the characteristics of the WAM.

*will be added*

– cloud cover at high (> 7 km), middle (2 km – 7 km) and low (< 2 km above ground) levels [%], 1-hourly

– column integrated water vapor [$\mathrm{kg\,m^{-2}}$], 3-hourly

385  – precipitation [mm per month], 3-hourly

– 2m temperature [K], 3-hourly

– 2m dew point temperature [K], 3-hourly

– mean sea level pressure [Pa], 3-hourly

– u- and v-wind at pressure levels 200 hPa and 600 hPa [$\mathrm{m\,s^{-1}}$], 3-hourly
* * *
**- Section 2.4:** Please give the units for each of the QoIs.

*will be added*

**Accumulated precipitation** [mm per month]: The accumulated precipitation fields are computed and averaged over the study region to obtain one scalar value representing the overall precipitation.

**Precipitation latitude** [° N]: The latitude of the rain belt is determined to investigate the potential influence of model

415  parameters on a north-south shift of the average precipitation. For this purpose, the latitudinal center of the accumulated pre-

**- Line 364:** '…all QoIs are averaged over the study period…' Please give more detail and clarity on the averaging periods/resolution (here, and/or with the individual QoI's below). How are they averaged? – Daily? 6-hourly?

*We will add more detail here and make clear which time resolution is used for averaging.*

characteristics of the WAM  are listed below. The temporal resolution of the recorded data is specified with the outputs. Notably, a finer resolution is applied to cloud cover data to accurately capture its anticipated higher variability.

- cloud cover at high (> 7 km), middle (2 km – 7 km) and low (< 2 km above ground) levels [%], 1-hourly
- column integrated water vapor [$kg\,m^{-2}$], 3-hourly
385   - precipitation [mm per month], 3-hourly
- 2m temperature [K], 3-hourly
- 2m dew point temperature [K], 3-hourly
- mean sea level pressure [Pa], 3-hourly
- u- and v-wind at pressure levels 200 hPa and 600 hPa [$m\,s^{-1}$], 3-hourly

from the ICON model output. The results for all QoIs are averaged over the study time (01 to 31 August of the years 2016,
410   2017, 2018 and 2019)  using all data with the temporal resolution given in Sect. 2.3.2. A schematic illustration of the monsoon
* * *
**- Line 371:** '…the longitudinal range is chosen…' Is this a fixed longitudinal range that is the same for all simulations?

*Yes, this is chosen for all simulation outputs, as the topography is the same, and to make the results comparable.*

of model parameters on the precipitation distribution becomes more evident. The range is fixed for all simulations to ensure
420   comparability of the results.
* * *
**- Line 477:** I think it might be useful to give a full definition of the parameter names on their first use in this section for the general reader, as they are not obvious from the acronyms.

*will be added*

Sensitivities of cloud cover (leftmost columns in Fig. 4) are generally dominated by the two deep-cloud parameters, the entrainment rate (*entrorg* ) and the terminal fall velocity of ice (*zvz0i*). High-level clouds are strongly affected by entrainsubstantial effect at these altitudes. Particularly, the  relative humidity threshold for onset of evaporation below the cloud base  (*rhebc_land_trop*) and the surface area density of the evaporative soil surface
545   (*c_soil* ), dominate the influence on low clouds, whereas deep-cloud parameters only play

**- Figure 6:** The labelling '1), 2),…' is difficult to see, especially when under dark shading. Could the numbering not be included with the names on the left/right for better clarity?

*will be clarified. We tried to use a consistent layout/labelling with the following figures, but we will make this clearer.*

[Figure]

**- Lines 515:** '…all other model parameters are set to their mean values…' Why is the mean value used for this choice? And not, say, the model's default values? How does this fixed choice for other parameters affect the results shown in Figure 5?

*The mean values correspond to the default values as the PDFs are defined that way. In L513-514 we emphasized that this illustration is meaningful - i.e. it is expected to be similar (only having vertical shifts) if the other parameters are chosen differently. We will elaborate this in more detail.*

such illustrations are meaningful. Figure 5 shows the individual relationships between each model parameter and each QoI, respectively, while all other model parameters are set to their mean values ‐which correspond to the ICON default values. Due
580 to the low parameter interactions, choosing different fixed values other than the mean values would predominantly result in vertical displacements of the presented curves in our analysis. The prediction variance from the Gaussian process regression (Eq. (4)) is indicated by the shaded areas around the curves.

**- Lines 522-523:** Here and elsewhere (including the Fig 7, Fig 8 and Fig 9 captions) I am very concerned about, and do not agree with, the interpretation of 'p-values' for the Kruskal-Wallis testing. For p-values, the general rules from basic statistics are that a p-value, $p \geq 0.05$ shows no evidence against the null hypothesis, H0, being tested, that $0.01 < p \leq 0.05$ indicates weak evidence against H0, that $0.001 < p \leq 0.01$ indicates strong evidence against H0, and then $p \leq 0.001$ is very strong evidence against H0. Hence, to say that $0.05 < p < 0.1$ shows high significance, and $p < 0.05$ shows very high significance is just clearly misleading. Please update the results and figures to have an appropriate interpretation of the p-values.

*Thank you for this comment. We will adjust the interpretation of the statistical test. The terms "very high significance" and "high significance" are misleading. We prefer to use levels*

*(in percentage) rather than the chosen labels. The interpretation will then need some adaptation.*

[Figure]

Results from the statistical Kruskal–Wallis test for the variability fields are denoted in Figs. 7, 8 and 9. Variability fields are denoted with two asterisks (∗∗) for  average p-values $p < 0.05$ (statistical significance on a 5 % level) and with one asterisk (∗) for  average p-values $0.05 < p < 0.10$  (statistical

590 significance on a 10 % level). While a 5 % significance level is a common choice, Quinn and Keough (2002) suggest that this threshold should not be rigid and should depend on specific circumstances. For instance, a larger sample size is more likely to yield statistically significant results. Given that our analysis includes only 60 training points, it is considered reasonable to include a less stringent significance level (10 %) as well. However, care must be taken to avoid overconfident statements. Overall, the validation results should be taken into account in the interpretation of local influences of the model parameters.

Figure 7. Spatial variability of selected output fields for the uncertain model parameters *entrorg* and *zvz0i*. The difference of the output quantity with respect to an increase of the model parameter value based on the sets $\mathbf{X}_{i,low}$ and $\mathbf{X}_{i,high}$ (see Sect. 2.5) is shown. Results of the statistical test are denoted with (∗∗):  statistical significance on a 5 % level,  (∗):  statistical significance   on a 10 % level. All scales are linear.
* * *
- As another example for the use of Universal Kriging in the Atmospheric Sciences, the following publication has recently employed it to obtain counter-factuals for shipping pollution: Diamond, M. S., Director, H. M., Eastman, R., Possner, A., & Wood, R. (2020). Substantial Cloud Brightening from Shipping in Subtropical Low Clouds. AGU Advances, 1, e2019AV000111. https:/doi.org/ 10.1029/2019AV000111

*See above comment (L114): We will include this in our literature review.*

115 meteorology, universal kriging has been applied in  several studies such as by  Glassmeier et al. (2019); Wellmann et al. (2020); Diamond et al. (2020), where either linear or quadratic basis functions are used as trend functions. However, to the authors' best knowledge, universal kriging with explicit *nonlinear* basis functions other than polynomials has not been applied in connection with meteorological applications. Furthermore, there have not been many studies regarding criteria for the choice of basis functions for universal

120 kriging.
* * *
- Just as a suggestion, I wonder whether some readers, notably those familiar with Gaussian Process emulators with Leeds involvement, might find it easier to relate to Section 2.2.2 if function choices were contrasted to those used in this literature, which to my knowledge, e.g., often assumes a Matérn co-variance structure, and would refer to the "aleatoric uncertainty due to weather noise" as "nugget effect".

*We agree that using other covariance functions such as the Matérn covariance is often meaningful. Even though the squared exponential function worked very well in our case, we will now mention the Matérn function as an established alternative choice. Also, we will refer to the 'nugget effect' to make the explanation more accessible to readers from different communities.*

> substantially between the parameters.  In addition to the radial basis function, the Matérn kernel function has also proven effective in the literature, particularly for its ability to better capture sharp jumps (Rasmussen and Williams, 2005)
> 270 . In our work, the radial basis function has proven to be practical and sufficient.

> 285 Here, additive i. i. d. Gaussian noise with variance $\sigma_n^2$ is considered, where $\sigma_n^2$ is treated as another hyperparameter to allow for aleatoric uncertainties, i. e. uncertainties that are attributed to weather noise in the ICON simulations.  In reference to the geostatistical origin of the method, this corresponds to the nugget effect (Matheron, 1969). The hyperparameters $\theta$ and $\sigma_n^2$ are determined by maximum likelihood estimation (Rasmussen and Williams, 2005). The noise level $\sigma_n^2$ can then provide an insight into aleatoric uncertainties in the ICON model. To speed up optimization, the gradient of the log marginal
* * *
- I would ask the authors to discuss further why there is so little interaction between the parameters. After all, the quantification of such interactions is a key strength of their approach. Could this be a consequence of the domain expertise that went into the selection of the 6 parameters?

*When selecting the 6 parameters, we aimed to include various effects on the WAM system, but we did not expect the parameter interactions to be so little. We would expect the interactions to be larger if we broadened the parameter ranges (PDFs).*

> units. Overall, the main and total effect indices do not differ strongly, which indicates that interactions between the parameters
> 535 are relatively weak. This justifies interpreting influences on QoIs of individual model parameters separately as done in this study. The interactions between the parameters is expected to be larger for broader parameter ranges as nonlinear effects may become more dominant.
* * *
- In how far are the below-cloud parameters related to cold-pool dynamics? Does their weak control on WAM characteristics imply anything for the relevance of parameterizing cold pools?

*The below-cloud parameters control how much rain is evaporated underneath the clouds and therefore modify surface rain and thermodynamic profiles. More evaporation creates cooler subcloud layers, which in turn leads to a larger negative buoyancy relative to neighboring grid cells and thus a larger lateral acceleration. This has some resemblance with having stronger cold pools but probably the grid spacing we use in our experiments (13 km) is not fine enough to fully resolve this process, including the triggering of new storms through cold pools. Nevertheless, the results we find for these parameters give some first indication about the general relevance of cold pools for the monsoon system and thus the potential gain from*

*a cold pool parameterization, which would attempt to represent the subgrid aspects of the problem.*

> 665 Increased evaporation leads to cooler subcloud layers, resulting in greater negative buoyancy and enhanced lateral acceleration compared to adjacent grid cells, somewhat akin to intensified cold pools. However, the 13 km grid spacing in our experiments
>
> may not adequately resolve this process, including new storm triggering by cold pools. Our findings therefore provide only limited insights into the actual significance of cold pools in the monsoon system and the potential benefits of a cold pool parametrization.
* * *
- The choice of using a 4-year "climatology" seems an important one, especially since emulators are cross-validated, and not tested on unseen data (i.e. an unseen 4-year period). Even though I would be surprised if the overall results would depend on this choice, some further elaboration would be helpful.

*See answer above to "Line 330"*

> 370 models. Preliminary studies with only one August month were still relatively volatile in a sense of large uncertainties in the surrogate model, which could then be strongly reduced by including four months. Validation of the surrogate model is essential to ensure that averaging over four years does not lead to non-realistic behavior in the average. The results of this validation should demonstrate a robust signal, indicating that the process results in a smoothing of individual signals rather than their cancellation. Using data from four different years also has the advantage of representing different states of SSTs, which are
* * *
- Section 2.5 was not detailed enough for me to fully grasp how the spatially resolved results were obtained.

*The explanation is indeed quite theoretical. We will add some detail to make it more intuitive.*

> of uncertain model parameters on the geographical distribution of the chosen output quantities (Sect. 2.3.2). For this purpose, the same all training points from the experimental design (Sect. 2.2.1) is are used, but for each model parameter $i$ the training points with the 25 % lowest and the 25 % highest values $x_i$ are selected. Let these sets for each model parameter $i$ be $X_{i,\text{low}}$ and
> 465 $X_{i,\text{high}}$. For example, the set $X_{\text{entrorg,low}}$ includes the 15 training points (25 % of 60 training points) with the lowest values of *entrorg* within the experimental design etc. For each training point, an ICON model simulation has been conducted and output fields are available. These output fields are averaged over the whole evaluation period (four August months). Furthermore, they are then averaged over the sets $X_{i,\text{low}}$ and $X_{i,\text{high}}$, respectively. As a result, the average spatial output data are averaged obtained for low and for high values of the considered uncertain model parametersseparately. Finally, these two averaged fields
> 470 . For example, using the set $X_{\text{entrorg,low}}$, all average output fields for low entrorg values are computed. Finally, the averaged fields with low ($X_{i,\text{low}}$) and high ($X_{i,\text{high}}$) values are subtracted to obtain a spatial variability field. The variability plot indicates in which regions the output value becomes higher or lower for an increase of model parameter $i$.

- L459ff: Isn't the aleatoric uncertainty sigma_n quantified?

*Yes, it is determined by maximizing the likelihood (as all hyperparameters) and then gives insight about the aleatoric uncertainty of the surrogate model. We will explain this in more detail.*

515  The standard deviation $\sigma_n$ of the  Gaussian noise in the regression model (see Sect. 2.2.2) provides an estimate of the aleatoric uncertainty in the ICON data. The results from the maximum likelihood estimation are also given in Table 3. Given that the values of $\sigma_n$ are generally lower but still of a similar magnitude compared to the RMSE, this indicates that a significant proportion of the

520 observed validation errors may be ascribed to aleatory uncertainties inherent in the weather model. The error attributed to the uncertainty in the surrogate model  is already relatively low, but could be further reduced by including more training points or averaging over more data (i. e. more years). A small RMSE (or NMSE) indicates that

**Table 3.** Validation results for the surrogate models of all QoIs with cross-validation.

| Quantity of Interest (QoI) | RMSE | $\sigma_n$ | unit | NMSE |
|---|---|---|---|---|
| cloud cover (high) | 0.862 | 0.247 | % | 1.22 % |
| cloud cover (mid) | 0.193 | 0.106 | % | 1.71 % |
| cloud cover (low) | 0.223 | 0.128 | % | 4.73 % |
| column integrated water vapor | 0.125 | 0.071 | $\mathrm{kg\,m^{-2}}$ | 7.04 % |
| temperature (2 m) | 0.054 | 0.025 | K | 3.88 % |
| dew point temp. (2 m) | 0.052 | 0.030 | K | 5.05 % |
| accumulated precipitation | 1.365 | 0.385 | mm per month | 11.64 % |
| AEJ speed | 0.069 | 0.088 | $\mathrm{m\,s^{-1}}$ | 55.24 % |
| TEJ speed | 0.116 | 0.059 | $\mathrm{m\,s^{-1}}$ | 7.02 % |
| ITD latitude | 0.100 | 0.055 | ° | 7.23 % |
| SHL latitude | 0.145 | 0.095 | ° | 4.75 % |
| AEJ latitude | 0.078 | 0.053 | ° | 5.81 % |
| precipitation center latitude | 0.057 | 0.049 | ° | 28.98 % |
| TEJ latitude | 0.044 | 0.022 | ° | 17.62 % |
| SHL pressure | 5.392 | 4.061 | Pa | 9.62 % |

- L382f: the "factorization" strategy needs elaboration.

*will be clarified*

of maximum zonal wind speeds. First, we compute the averaged zonal wind speeds for each latitude on the grid.

430  This step simplifies the data to a manageable form with one average wind speed value for each latitude on the grid. The distribution of these wind speeds is still relatively flat making it difficult to robustly determine the latitude of maximum wind speed. We thus employ a strategy where we exponentiate the average wind values (here,  an exponent of 3 yielded useful results) to assign higher weight to the highest values and to reduce the influence of values far from the jet center which are still relatively high (Figs. 6.5 and 6.6). Finally, we determine the weighted average of latitudes analogously

435 as for the precipitation center (Fig. 3b). Without the exponentiation strategy the chosen latitudinal range for the analysis would have a major effect on the result, which should be avoided.

- L149: description of tkhmin needs more detail.

*More detail will be added with another reference to literature of the DWD.*

implication, namely deep-cloud (*entrorg*, *zvz0i*), below-cloud (*rhebc_land_trop*, *rcucov_trop*) and boundary-layer (The choice of *tkhmin* , *c_soil*) parameters.

Despite the different physical influence of influences the turbulent diffusion of heat and moisture, which can in some situations impact cloud formation. Some level of vertical diffusion is practically always present in reality. In case of highly stable conditions and weak vertical wind shear, however, this diffusion is underestimated by the turbulence parametrization. Therefore a minimum vertical diffusion (i. e., *tkhmin*) is set in the model to counteracts this underestimation (Raschendorfer, 2012) . The parameter *c_soil* denotes the evaporating fraction of soil in the form of a unitless fraction. Higher values lead to higher

---

## Author Response (AR2)

**Point-by-point response to referee comments on revised submission**

**Quantifying uncertainty in simulations of the West African Monsoon with the use of surrogate models**

*Matthias Fischer, Peter Knippertz, Roderick van der Linden, Alexander Lemburg, Gregor Pante, Carsten Proppe, John H. Marsham*

We would like to thank the reviewer for the final remarks and we want to comment on those. Technical remarks are corrected.
* * *
**- Section 2.1.1, Para1: Lines 203-205:** 'Since probability varies strongly across the input space, it is meaningful to train the model with higher accuracy in regions with higher probability. This is because we construct surrogate models particularly for performing global sensitivity analysis.'
**- Lines 206-207:** 'Thus, for a more accurate sensitivity analysis, it is crucial for the model to exhibit higher accuracy in regions of the parameter space where the PDF values are greater.'

I think these statements are in some ways misleading and this paragraph should be edited to remove any misconceptions / ensure clarity and to also acknowledge the potential negative consequences of the sampling strategy applied. Following the author's response to my previous comment on the sampling for the training data (on training the model with higher accuracy in regions with higher probability, previously lines 195-196), I'm still not convinced about the validity of this. I can understand that the authors want the surrogate to be as accurate as possible where they will sample more, but with a fixed amount of training data in total (as I think is the case here – using $n = 10*p$ (where p is the number of parameters perturbed), this approach must have an opposite effect on the accuracy of the samples in areas of lower probability (by reducing it), which although sampled less, **can/will still be sampled in a global sensitivity analysis and so can affect the results of it**. There is **no evidence** provided (or to my knowledge) to say that this approach / sampling strategy is **crucial** to obtain a more accurate sensitivity analysis, and I think it is just as possible that it could lead to less accurate sensitivity results.

The fact that the authors intend to perform a global sensitivity analysis doesn't make sense to me as a reason to vary the accuracy of the underlying model (here, the emulator/surrogate model) that you want to understand the sensitivity of. The effects of the PDFs are still accounted for in the sampling of the sensitivity analysis procedure itself, and so this seems to be an unnecessary step that has potential to induce possibly significant inaccuracy in some emulator predictions and hence the obtained sensitivity results. When constructing a surrogate model, technical aspects such as changes in the smoothness of the surface that one is trying to approximate can affect the emulators accuracy around the input space and so be valid reasons for the requirement of more/less training data in different areas – In my experience, if more data is needed, this is added in addition to the base training sample of size $10*p$. Given this, it seems also possible that the outcome of the sampling strategy described could lead to fewer training points in areas of input space that the Gaussian process might already find the output more difficult to capture well [if they happen to be the areas of lower probability], which would then further lead to poor representation of the climate model, which could affect sensitivity results.

I understand that it isn't possible (due to computational expense) to re-run the study with a uniform training sample for the surrogate model and do the direct comparison, and also that validation of the surrogate models should provide some evidence that the emulator prediction is reasonable across input space [this evidence seems limited here in showing prediction accuracy in different areas of input space]. However, I think it is important that any caveats of the sampling strategy used are

clearly acknowledged [i.e. that the global sensitivity analysis **can/will** still sample in areas of low probability, where the emulator here will be less accurate, which could adversely affect the resulting sensitivities] and that all statements of something being 'better' or 'crucial' are either evidenced or not used.

*Thank you for the very detailed comment. We have clarified the limitations of this methodology in the manuscript so that incorrect conclusions are avoided. In particular, the argument that an experimental design with a density of training points equal to the probability density function (i.e. higher density in more probable regions) would be optimal for a global sensitivity analysis (GSA) has been weakened as this requires further methodological investigations that cannot be carried out at this point. We highlighted that regions with a sparse distribution of training points (i.e. the tails of the PDFs) can strongly influence the outcome of a global sensitivity analysis, even though they are sampled less frequently.*

*Furthermore, we added the possibility to add further training points if needed by sequential sampling techniques depending on the model accuracy.*

*Finally, we emphasize that the explained methodical step should be skipped, if a surrogate model with equal probability within predefined input parameter ranges was desired. In that case a uniform density of training points (e.g. a standard Latin hypercube design) may be used.*
* * *
200 **2.2.1 Training points**

In order to build a surrogate model, training points for the model parameters have to be defined based on the PDFs specified in Sect. 2.1. Hereafter, we will refer to the model parameter space as *input space*, as commonly done in the scientific discipline of Uncertainty Quantification (UQ). Since probability varies  substantially across the input space,  the density of points selected in the parameter space for global sensitivity analysis corresponds to the probability density function (PDF), resulting

205 in regions of higher probability being sampled more frequently. Therefore, it is considered meaningful to train the model with higher accuracy in  these regions. However, this method inherently leads to a reduced focus on areas of lower probability, which, despite being sampled less frequently, are still essential for a comprehensive global sensitivity analysis. The   assumption that prioritizing areas of

210 higher probability leads to more accurate sensitivity analysis outcomes requires further scientific investigation. Furthermore, sequential algorithms can be employed to supplement the base design with additional training points in regions where enhanced model accuracy is required. Additionally,   if the  sole objective were to develop a surrogate model with  uniform accuracy across the  entire parameter space, including the tails of the PDFs, then

215 employing a uniform density of training points would be more  appropriate. In our case, using more training points in regions with higher probability leads to an experimental design with inhomogeneous space-filling properties where surrogate modeling methods may struggle. As a consequence, the trained surrogate models may have problems to predict QoIs in the tails of the PDFs. Therefore, we transform the *physical* (hereinafter used to denote parameter PDFs according to Table 1) input space to an independent and identically distributed (i. i. d.) uniform input space. In the transformed uniform input space, which

220 can be thought of as a multidimensional unit hypercube, every region is associated with equal probability and thus we can apply a space-filling sampling technique. In particular, we use maximin Latin hypercube sampling (Morris and Mitchell, 1995) to define 60 training points. We use the recommendation given by Loeppky et al. (2009) for choosing the number of training points as $n = 10p$, where $p$ is the number of input dimensions ($p = 6$ in our case).